

# Insights into the phylogenetic relationships and species boundaries of the *Myricaria squamosa* complex (Tamaricaceae) based on the complete chloroplast genome

Huan Hu[1,*], Qian Wang[1,*], Guoqian Hao[2], Ruitao Zhou[3], Dousheng Luo[3], Kejun Cao[3], Zhimeng Yan[4] and Xinyu Wang[5,6]

[1] Microbial Resources and Drug Development Key Laboratory of Guizhou Tertiary Institution, Zunyi Medical University, Zunyi, China
[2] School of Life Science and Food Engineering, Yibin University, Yibin, China
[3] School of Preclinical Medicine, Zunyi Medical University, Zunyi, China
[4] School of Medical Information Engineering, Zunyi Medical University, Zunyi, China
[5] Key Laboratory of Medical Electrophysiology, Institute of Cardiovascular Research, Southwest Medical University, Luzhou, China
[6] Department of Cardiology, The Affiliated Hospital of Southwest Medical University, Luzhou, China
* These authors contributed equally to this work.

Corresponding authors
Huan Hu, huhuan1990@163.com
Xinyu Wang, hellowangxy@swmu.edu.cn

## ABSTRACT

*Myricaria* plants are widely distributed in Eurasia and are helpful for windbreak and embankment protection. Current molecular evidence has led to controversy regarding species boundaries within the *Myricaria* genus and interspecific phylogenetic relationships between three specific species—*M. bracteata*, *M. paniculata* and *M. squamosa*—which have remained unresolved. This study treated these three unresolved taxa as a species complex, named the *M. squamosa* complex. The genome skimming approach was used to determine 35 complete plastome sequences and nuclear ribosomal DNA sequences for the said complex and other closely related species, followed by *de novo* assembly. Comparative analyses were conducted across *Myricaria* to identify the genome size, gene content, repeat type and number, SSR (simple sequence repeat) abundance, and codon usage bias of chloroplast genomes. Tree-based species delimitation results indicated that *M. bracteata*, *M. paniculata* and *M. squamosa* could not be distinguished and formed two monophyletic lineages (P1 and P2) that were clustered together. Compared to plastome-based species delimitation, the standard nuclear DNA barcode had the lowest species resolution, and the standard chloroplast DNA barcode and group-specific barcodes delimitated a maximum of four out of the five species. Plastid phylogenomics analyses indicated that the monophyletic *M. squamosa* complex is comprised of two evolutionarily significant units: one in the western Tarim Basin and the other in the eastern Qinghai-Tibet Plateau. This finding contradicts previous species discrimination and promotes the urgent need for taxonomic revision of the threatened genus *Myricaria*. Dense sampling and plastid genomes will be essential in this effort. The super-barcodes and specific barcode candidates outlined in this study will aid in further studies of evolutionary history.

# INTRODUCTION

*Myricaria* Desv. is one of the three genera in the Tamaricaceae family, which was established by *Desvaux (1825)* by dividing certain species from the genus *Tamarix* L. (*Zhang & Zhang, 1984*). *Myricaria* contains about 13 described species, mainly distributed in the Qinghai-Tibet Plateau (QTP) and adjacent north-temperate areas of the Eurasian continent (*Yang & Gaskin, 2007*; *Wang et al., 2009*). The dominant species, *M. germanica* (L.) Desv., as well as two local species, *M. dahurica* (Willd.) Ehrenb. and *M. longifolia* (Willd.) Enrenb. grow in central Asia and western Europe. The remaining ten species, of which four are endemic to China—*M. pulcherrima* Batal., *M. wardii* Marquand, *M. laxiflora* (Franch.) P. Y. Zhang & Y. J. Zhang and *M. paniculata* P. Y. Zhang & Y. J. Zhang (*Wang et al., 2006*; *Liu, Wang & Huang, 2009*; *Wang et al., 2009*)—are naturally distributed in the montane areas of western and northern China, close to rivers or lakes. The latest attempt at taxonomic revision of *Myricaria* species based on only morphological data did not yield satisfactory results (*Zhang & Zhang, 1984*). Recent extensive field investigations have revealed that the taxonomic status of *M. squamosa* Desv., *M. bracteata* Royle and *M. paniculata* is ambiguous and problematic (*Wang et al., 2006*). These three closely related species have widespread distributions in and around China, and their morphological differences are not always clear and consistent. In this study, *M. squamosa*, *M. bracteata* and *M. paniculata* were treated as a species complex, named the *M. squamosa* complex.

*M. squamosa* was the first taxon described by Desvaux in 1825 and is widely distributed in Central and East Asia from the Altai mountains to the Himalayas. *M. bracteata* was described by J.F. Royle in 1839 and is native to central and northern China, western Himalaya, the Pamirs, and the Tianshan, Sayan and Caucasus Mountains. In their 1984 revision, Zhang and Zhang classified specimens of *M. germanica* collected in China as a distinct species, naming it *M. paniculata*. This species is primarily found in regions ranging from Siberia to the Qinling and Hengduan Mountains (*Zhang & Zhang, 1984*; *Wang et al., 2006*). Numerous specimens and practical field investigations have shown that the racemes of *M. squamosa* are frequently lateral on ancient branches and fascicled in axils, and the inflorescences of *M. bracteata* and *M. paniculata* are usually terminal. *M. paniculata* has two inflorescence types: lateral racemose in spring on the branches of the previous year and paniculate terminal lax in summer and autumn on the current year's branches. *M. bracteata* only has a dense racemose terminal type of inflorescence on the branches of current year, with bracts that are broadly ovate and broader than both *M. squamosa* and *M. paniculata* (Figs. 1C–1E). Inflorescences on the branches of the previous year typically have numerous persistent imbricate scales at the base, while inflorescences on the current year's branches do not.

The initial step of taxonomic analyses traditionally involves delimiting groups of individuals based on their morphological resemblance and phenotypic distinctiveness.

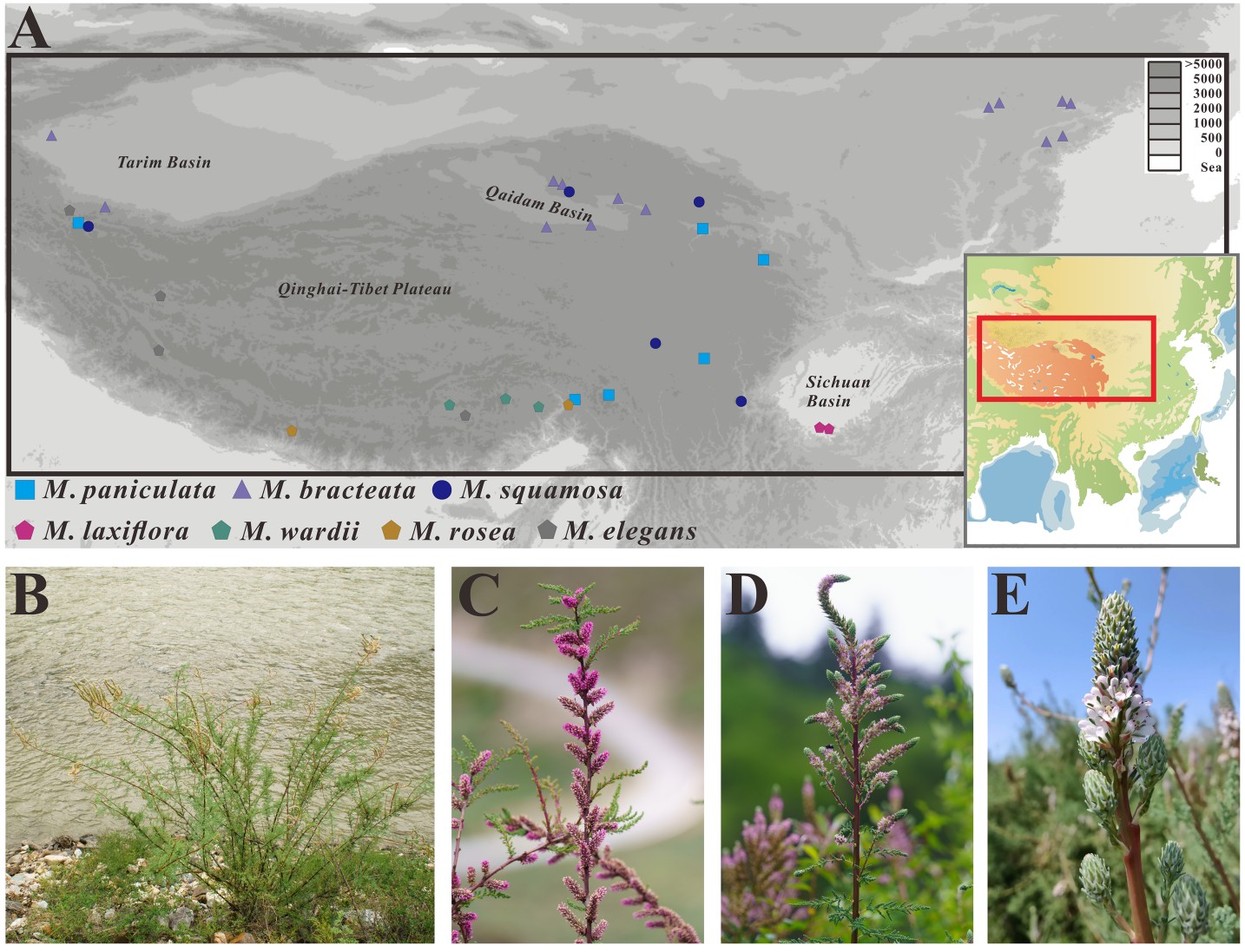

**Figure 1 Distribution map and morphological characteristics of the *Myricaria* species.** (A) Distribution map of the *Myricaria* samples collected in this study. (B) Morphology and habitat of *M. paniculata*. (C–E) Inflorescences of *M. squamosa*, *M. paniculata* and *M. bracteata*, respectively (Photos by Guoqian Hao and Huan Hu).

Although *M. squamosa*, *M. bracteata* and *M. paniculata* have slight morphological differences in inflorescence type, bract size and imbricate scales, there has been debate about whether these morphological differences are sufficient to justify the delineation of taxa (*Wang et al., 2009*). Previous studies on the phylogenetic relationships of *Myricaria* species have mainly focused on the specific taxonomic status of *M. elegans* Royle (*Zhang, Yin & Pan, 2001*; *Zhang et al., 2003*; *Hua, Zhang & Pan, 2004*).

Species delimitation is crucial in various biological disciplines, such as biogeography, ecology, and evolutionary biology (*Sites & Marshall, 2003*; *Reydon & Kunz, 2019*). Species serve as a metric in biology and are essential for conservation, as species help conservationists develop effective strategies for targeted conservation management (*Coates, Byrne & Moritz, 2018*). Accurately defining species allows the study of patterns of

genetic diversity and population structure within certain categories, as well as the factors driving speciation (*Nunes, Raxworthy & Pearson, 2022*). For example, the discovery and precise description of *Ostryopsis intermedia* B. Tian & J. Q. Liu sparked a series of research projects focused on its homoploid hybrid speciation. These findings provided insights into how Quaternary climate change triggered its demographic history and biogeography pattern, as well as the underlying genetic mechanism of its speciation (*Tian & Liu, 2010*; *Liu et al., 2014*; *Wang et al., 2021*). Similar studies have been conducted on species exhibiting hybrid speciation, such as *Allium przewalskianum* Regel, *Picea purpurea* Mast., and *Cupressus chengiana* S.Y. Hu (*Liang et al., 2014*; *Ru et al., 2018*; *Li et al., 2020*; *Wu et al., 2022*).

Since Ernst Mayr proposed the concept of species, it has sparked a heated debate and resulted in a variety of definitions (*Mayden, 1997*). The traditional morphological (taxonomic) species concept is based on discontinuous phenotypic differences across species at the population level (*Cronquist, 1978*; *Saraswati & Srinivasan, 2016*), but it can be inaccurate when cryptic species, phenotypic polymorphism, or adaptive convergence are involved. Advances in molecular biology and phylogenetic analytic improvements have led to increased popularity of the phylogenetic species concept, which refers to a species as an independent evolutionary lineage, and sidesteps the non-universal criterion of reproductive isolation, making it applicable to both extant and extinct organisms, as well as to both sexual and asexual reproductive creatures (*Mishler & Donoghue, 1982*; *Nixon & Wheeler, 1990*). As a result, DNA-based approaches have become increasingly important in resolving taxonomic uncertainties and identifying evolutionarily significant units (ESUs; *Li et al., 2020*).

In an era marked by widespread species extinction and ongoing environmental crises, DNA barcoding has emerged as a standardized tool for species delimitation due to its ability to facilitate accurate and direct comparison among different users (*Hebert, Ratnasingham & deWaard, 2003*; *Li et al., 2015*; *de Boer et al., 2017*; *Antil et al., 2023*). DNA barcoding is based on the principle of barcoding gap, which refers to the difference in genetic distances between intra- and inter-species. Previous studies have assessed different combinations of nuclear internal transcribed spacers (ITS) with the seven leading candidate plastid DNA regions (*Hollingsworth et al., 2009*) as potential core DNA barcodes discriminating closely related plant species (*Kress & Erickson, 2007*; *Cheng et al., 2021*). However, these combinations may not always provide adequate barcoding gaps and efficiency compared to the *CO*1 barcode used in animals, especially in plants with ancestral polymorphisms, recent speciation, or hybridization (*Parks, Cronn & Liston, 2009*; *Twyford, 2014*). Previous studies using traditional DNA barcodes for the *M. squamosa* complex showed poor species resolution due to insufficient specimen sampling and genetic diversity; *M. squamosa*, *M. bracteata* and *M. paniculata* shared haplotypes and tended to cluster into one clade (*Liu, Wang & Huang, 2009*; *Wang et al., 2009*). This highlights the need for more extensive taxon sampling and the development of more discerning super-barcodes to identify the phylogenetic relationships and remove the taxonomic ambiguity within this genus.

The chloroplast, a high-copy organelle in plants, is one of the most technically accessible regions of the plant genome. With the development of next-generation sequencing technologies and the reduction of sequencing costs, the plastome has gained recognition as a "super-barcode" with versatile applications in plant phylogenetics, species delimitation and population genetics, especially in groups with poor morphological differentiation among species (*Parks, Cronn & Liston, 2009*; *Yang et al., 2013*; *Li et al., 2015*; *Ruhsam et al., 2015*; *Liu et al., 2021*).

Although previous studies have analyzed plastome features with a few *Myricaria* species available in GenBank, there is currently no report on species-specific identification and determination of species boundaries within the *M. squamosa* complex (*Wang et al., 2020*; *Han et al., 2021*). Plastome sequences and genome skimming can also be used for nuclear ribosomal DNA (nrDNA) assembly, enabling the inference of phylogenetic relationships using both uniparentally-inherited plastomes and biparentally-inherited nuclear genes (*Wen et al., 2018*; *Liu et al., 2019*, *2020a*, *2021*).

To explore the features and structural differentiation of plastomes among the species in the *M. squamosa* complex, and to establish a well-supported phylogenetic framework of Chinese *Myricaria*, 25 individual samples, representing three recognized taxa in the *M. squamosa* complex through a dense taxon sampling strategy, were used in this study. Complete plastomes and nrDNA sequences *via* genome skimming were used to test their discriminatory power in the *M. squamosa* species complex, an evolutionarily young lineage where the traditional DNA barcoding approach has been insufficient. The main purposes of this study were to: (i) gain insight into the structural features of multiple *Myricaria* plastomes; (ii) determine whether plastome sequences can provide a more detailed resolution of the shallow-level relationships within the *M. squamosa* complex compared with nrDNA sequences; and (iii) evaluate the utility of proposed '*Myricaria*-specific' plastid barcodes for species discrimination.

# MATERIALS AND METHODS

## Sampling, DNA extraction, and sequencing

A total of 25 samples representing different populations of the *M. squamosa* species complex, as well as 10 individual samples of *M. laxiflora*, *M. wardii*, *M. elegans* and *M. rosea* W. W. Sm. were collected from southwest and northwest China from July to September, spanning years 2019 to 2021 (Fig. 1). Additionally, four previously-published *Myricaria* plastid genomes were obtained from NCBI. In total, seven recognized taxa of *Myricaria* were sampled, with each species having more than two populations. Data from *Tamarix* and *Reaumuria* L. were used as outgroups. *Myricaria* samples sequenced in the present study were not included in the list of national key protected plants and not collected from national parks or natural reserves. The research methods were approved by the Ethical Experimentation Committee of Zunyi Medical University (Identification Code: ZMU-BO-1903-169) and followed the legal and ethical standards of the local government. The formal identifications of all samples were undertaken by Professor Guoqian Hao (Yibin University) and Associate Professor Qian Wang (Zunyi Medical University) based on the most widely-used morphologic criterion (*Yang & Gaskin, 2007*). Voucher

specimens from each population were then deposited at the Herbarium of Zunyi Medical University Life Science Museum with specific voucher numbers. Detailed information of the species and datasets are listed in Table S1.

Total genomic DNA was extracted from approximately 200 mg of silica-dried leaves using a Magnetic Universal Genomic DNA Kit (DP705; TIANGEN, Beijing, China). Purified genomic DNA samples were sent to Novogene (Beijing, China) for next-generation sequencing on the Illumina NovaSeq 6000 platform, with a 2 × 150-bp paired-end run.

## Plastid genome and nrDNA assembly, annotation and visualization

All raw Illumina data, ~2 Gb of raw data for each sample, were filtered using Trimmomatic 0.39 (*Bolger, Lohse & Usadel, 2014*) under default parameters to remove adapters and low-quality bases. Clean reads were assembled by Oases 0.2.09 (*Schulz et al., 2012*) and Velvet 1.2.10 (*Zerbino & Birney, 2008*) on an array of single-*k* assemblies. These assemblies were merged into a final assembly and then mapped to the published *M. laxiflora* plastome (GenBank accession: MN867948) using Burrows-Wheeler Alignment tool (BWA) 0.7.17 (*Li & Durbin, 2009*) and SAMtools 1.10 (*Danecek et al., 2021*). The orders of aligned contigs were determined according to the reference genome by Geneious 8.1.4 (*Kearse et al., 2012*). The plastid genomes were annotated using Plastid Genome Annotator (PGA; *Qu et al., 2019*) and *Amborella trichopoda* Baill. (AJ506156) was as a reference, and then manually corrected with Geneious, based on the recommended reference. Chloroplot (*Zheng et al., 2020*) was used to illustrate a circular genome map.

The entire internal transcribed spacer sequence (ITS: ITS1, 5.8S, and ITS2) was concatenated to obtain nrDNA sequences for each sample, using *Tamarix chinensis* Lour. (KT377278) as a reference for assembly. This was done using a modified reference-based method and a reference similar to that of the plastomes (*Liu et al., 2020a*; *Su et al., 2021*). The clean reads were mapped to the reference using Bowtie2 2.4.5 (*Langmead & Salzberg, 2012*) and SAMtools 1.10 (*Danecek et al., 2021*), resulting in a BAM file with only mapped reads. The BAM file was then imported into Geneious 8.1.4 and consensus sequences were extracted to serve as the final nrDNA sequences.

### *Myricaria* plastome feature analysis

For a comprehensive understanding of the *Myricaria* plastome features, six individuals representing six lineages from the complete plastome phylogenetic result were selected for further comparison.

### Codon usage bias and gene selective pressure analysis

The codon usage bias parameters, containing effective number of codons (ENC), relative synonymous codon usage (RSCU) value and GC content at the first, second, third base and third synonymous position (GC1, GC2, GC3, and GC3s), were estimated using CUSP and CHIPS plugins in EMBOSS (*Lamprecht et al., 2011*) and the CodonW 1.4.2 program (https://codonw.sourceforge.net/). These analyses were performed on protein-coding genes (PCGs) larger than 300 bp in size.

To accurately detect site-specific positive selection in the protein-coding sequences (CDSs) of *Myricaria*, the orthologous PCGs were extracted from six *Myricaria* plastomes by a custom Perl script, and aligned using MUSCLE v5 (*Edgar, 2021*). *M. laxiflora* was then compared with the other five individuals in 80 shared unique PCGs to analyze the Ka and Ks substitution rates and Ka/Ks ratio using KaKs Calculator 2 (*Wang et al., 2010*) set to genetic code table 11 and the $\gamma$-NG method of calculation (*Yang et al., 2023*). The selective pressure of each gene was predicted by considering the ratio of Ka/Ks: Ka/Ks < 1 was identified as purified selection, Ka/Ks = 1 was neutral selection and Ka/Ks > 1 was identified as positive selection (*Yang & Nielsen, 2000*).

### Repeat sequences analysis

Simple sequence repeats (SSRs) loci were searched in MISA-web (*Beier et al., 2017*), with the threshold value of repeat number as ≥10 for mono-nucleotide repeats, ≥5 for di-nucleotide repeats, ≥4 for tri-nucleotide repeats, and ≥3 for tetra-, penta-, and hexa-nucleotide repeats. The maximum sequence length between two SSRs to create a compound SSR was 100 bp (*Yang et al., 2023*). Tandem repeats (TRs) were detected by analyzing the plastome sequences in TRF software with the parameters recommended by the official manual: 2, 7, 7, 80, 10, 50, 500, -f, -d, -m (*Benson, 1999*). Four types of long repetitive sequences (forward, reverse, complement and palindromic) were detected using the online REPuter program with the maximum repeat size being set at 50 bp, the minimum repeat size set to 30 bp, and hamming distance set to 3 (*Kurtz et al., 2001*). All the repeats identified by these three programs were manually verified to remove redundant and nested results. It is important to note that repeats located in the IR regions were counted only once.

### Species-specific DNA barcode development

Nucleotide diversity (*P*i) of the *Myricaria* plastid genomes was calculated with a sliding window analysis using DnaSP v.6.10 (window length = 800 bp and step size = 200 bp; *Rozas et al., 2017*). Regions with relatively high *P*i were defined as hyper-variable regions, and tended to also have high species resolution. The discrimination power of specific barcodes was inferred by a tree-based method using the maximum likelihood (ML) analysis in IQ-TREE web server (*Trifinopoulos et al., 2016*).

### Chloroplast genome comparison within Myricaria

The web-based mVISTA server (*Frazer et al., 2004*) was used to identify sequence and structural variations among *Myricaria* species under Shuffle-LAGAN mode, using *M. laxiflora* as the reference. IRscope was employed to compare and visualize the LSC/IRb and SSC/IRa boundaries (*Amiryousefi, Hyvönen & Poczai, 2018*). Evolutionary divergence (*p*-distances) among the 39 *Myricaria* accessions were evaluated using MEGA 11 (*Tamura, Stecher & Kumar, 2021*).

## Phylogenetic profiles and species boundary test

Only cases where more than one individual was sampled per species were used in the species boundaries assessment in this study. It is worth noting that the published plastome

of *M. prostrata* J. D. Hooker & Thomson (MN088847) was excluded from our analysis. This prevented singleton species samples from occupying phylogenetic space and 'disrupting' species-level monophyly, leading to 'failed' species recovery (*Wang et al., 2022a*).

Four datasets were independently used for species delimitation by a tree-based method: (1) the complete plastome sequences, (2) standard nrDNA barcodes (ITS), (3) the plastid DNA barcodes (*mat*K+*rbc*L+*trn*H-*psb*A+*trn*L-F) recommended by *Li et al. (2015)* and (4) the group-specific DNA barcodes selected from hyper-variable regions in plastomes. Each dataset containing outgroups was aligned using the MAFFT online service with default parameters (*Katoh, Rozewicki & Yamada, 2019*) and manually checked using MEGA 7.0 (*Kumar, Stecher & Tamura, 2016*). The best-fit substitution model for each dataset was then determined using Jmodeltest 2.1 (*Guindon & Gascuel, 2003*; *Darriba et al., 2012*). ML phylogenetic analyses were then performed through the IQ-TREE web server with 1,000 bootstrap replicates using UFBoot2 and the collapsing near-zero branches option (*Trifinopoulos et al., 2016*; *Hoang et al., 2018*). Bayesian inference (BI) analyses were inferred using MrBayes v3.2.7 with two simultaneous parallel analyses employing the MC3 algorithm (*Ronquist et al., 2012*). For each run, four independent Markov Chain Monte Carlo (MCMC) chains (one cold and three heated) were propagated for 1,000,000 generations each and sampled every 500 generations. The log file was checked to ensure that the potential scale reduction factor (PSRF) was reasonably close to 1.0 and the estimated sample size (ESS) was larger than 200 for all parameters. The first 25% of samples were conservatively discarded as burn-in. The remaining trees were used to generate a 50% majority-rule consensus tree. The ML and BI trees were annotated and visualized using ITOL 6.5.8 (*Letunic & Bork, 2021*).

To check for discrepancies between different ML trees derived from nrITS and plastome datasets, the ConsensusNetwork algorithm in SplitsTree v4.19.1 (*Huson & Bryant, 2006*) was used to compute the consensus splits of the two trees to produce a consensus network.

## RESULTS

### Characteristics of newly-obtained *Myricaria* plastomes

The *Myricaria* plastomes ranged in size from 154,485 to 155,347 bp in length, and exhibited a quadripartite structure (Fig. 2), including a large single-copy region (LSC, 84,216–84,825 bp), a small single-copy region (SSC, 18,238–18,319 bp), and two inverted repeated regions (IRs, 25,966–26,149 bp; Table S1). The total GC content was 36.3–36.6%, with the IRs having a higher GC content (42.4–42.5%) compared to LSC (34.0–34.1%) and SSC (29.5–29.9%). The *Myricaria* plastomes encoded a total of 88–90 PCGs (80–81 PCG species), 8 rRNAs (4 rRNA species) and 37 tRNAs (30 tRNA species; Fig. 2). A total of 19–20 genes were duplicated in the IRs, including 4 rRNAs (*rrn*16, *rrn*23, *rrn*4.5 and *rrn*5), 7 tRNAs (*trn*I$^{CAU}$, *trn*L$^{CAA}$, *trn*V$^{GAC}$, *trn*I$^{GAU}$, *trn*A$^{UGC}$, *trn*R$^{ACG}$ and *trn*N$^{GUU}$) and 8–9 PCG species (*rpl*2, *rpl*23, *ycf*2, *ycf*15, *ndh*B, *rps*12, *ycf*68, *rps*7 or the additional *ycf*15). Of the 11 genes with introns detected, nine had a single intron (*rps*16, *atp*F, *pet*B, *pet*D, *rpl*16, *rpl*2, *ndh*B, *rps*12, *ndh*A), while two had double introns (*ycf*3, *clp*P).

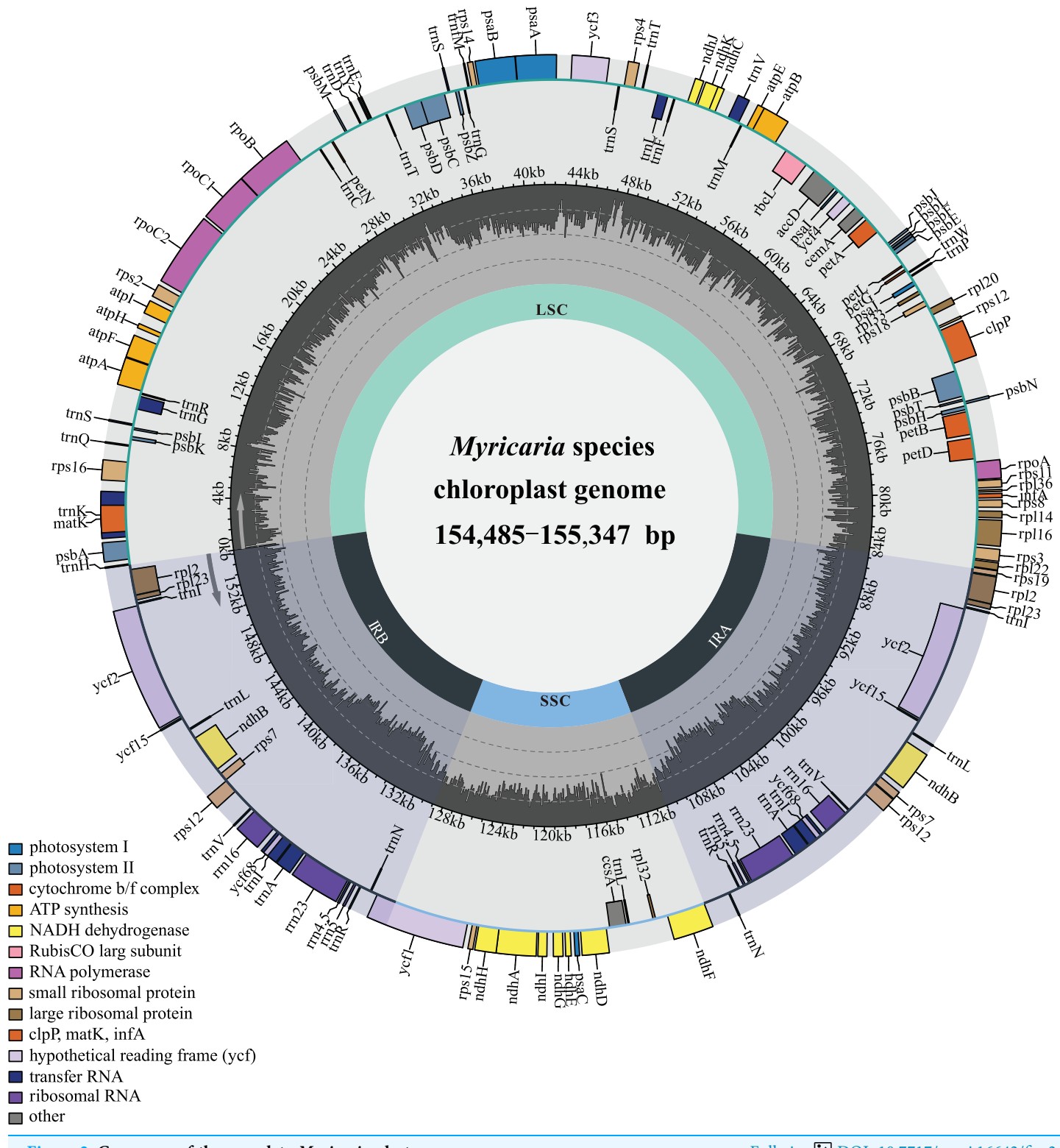

**Figure 2  Gene map of the complete *Myricaria* plastome.**

**Table 1 Gene contents in the chloroplast genomes of *Myricaria* species.**

| Category | Group of genes | Gene names | Amount |
|---|---|---|---|
| Photosynthesis | Photosystem I | *psa*A, *psa*B, *psa*C, *psa*I, *psa*J | 4 |
| | Photosystem II | *psb*A, *psb*B, *psb*C, *psb*D, *psb*E, *psb*F, *psb*H, *psb*I, *psb*J, *psb*K, *psb*L, *psb*M, *psb*N, *psb*T, *psb*Z | 15 |
| | NADH dehydrogenase | *ndh*A\*, *ndh*B\*(2), *ndh*C, *ndh*D, *ndh*E, *ndh*F, *ndh*G, *ndh*H, *ndh*I, *ndh*J, *ndh*K | 12 |
| | cytochrome b/f complex | *pet*A, *pet*B\*, *pet*D\*, *pet*G, *pet*L, *pet*N | 6 |
| | ATP synthase | *atp*A, *atp*B, *atp*E, *atp*F\*, *atp*H, *atp*I | 6 |
| | Large subunit of rubisco | *rbc*L | 1 |
| Self-replication | Proteins of large ribosomal subunit | *rpl*14, *rpl*16\*, *rpl*2\*(2), *rpl*20, *rpl*22, *rpl*23(2), *rpl*32, *rpl*33, *rpl*36 | 11 |
| | Proteins of small ribosomal subunit | *rps*11, *rps*12\*\*(2), *rps*14, *rps*15, *rps*16\*, *rps*18, *rps*19, *rps*2, *rps*3, *rps*4, *rps*7(2), *rps*8 | 14 |
| | Subunits of RNA polymerase | *rpo*A, *rpo*B, *rpo*C1\*, *rpo*C2 | 4 |
| | Ribosomal RNAs | *rrn*16(2), *rrn*23(2), *rrn*4.5(2), *rrn*5(2) | 8 |
| | Transfer RNAs | 37 tRNAs (6 contain an intron, 7 in the IRs) | 37 |
| Other genes | Maturase | *mat*K | 1 |
| | Protease | *clp*P\*\* | 1 |
| | Envelope membrane protein | *cem*A | 1 |
| | Acetyl-CoA carboxylase | *acc*D | 1 |
| | c-type cytochrome synthesis gene | *ccs*A | 1 |
| | Translation initiation factor | *inf*A | 1 |
| Genes of unknown function | Proteins of unknown function | *ycf*1, *ycf*15(2), *ycf*2(2), *ycf*3\*\*, *ycf*4, *ycf*68(2) | 8 |

Notes:
Gene\*, gene with one intron.
Gene\*\*, gene with two introns.
Gene(2), number of copies of multi-copy genes.

The genome size, GC content, gene number and gene order in the *Myricaria* plastomes were relatively conserved compared to the outgroups, with the exception of an early terminator observed in the *ycf*15 gene in a few individuals (Table S1). The major genes in the plastome of *Myricaria* species could be roughly divided into three functional categories (Table 1), with genes associated with photosynthesis and self-replication comprising the majority of the chloroplast genome.

### Codon usage bias and gene selective pressure analysis

The number of CDSs in the six newly-assembled *Myricaria* plastomes ranged from 50 (*M. squamosa* complex P2) to 51, based on a length threshold of 300 bp for codon preference analysis. The ENC values varied from 34.44 to 51.66, with the highest observed in the *rps*3 gene and lowest values in the *rps*16 and/or *rps*14 genes, which frequently had values smaller than 35. The overall GC content of the plastomes was consistent across all six samples, ranging from 36.91% to 37.06%. As expected, the GC1, GC2, GC3 and GC3s contents were also consistent among the plastomes (Table S2).

The number of codons in the CDSs of the six plastomes ranged from 20,520 (*M. rosea*) to 20,713 (*M. wardii*). Among all samples, leucine was the most abundant amino acid, with
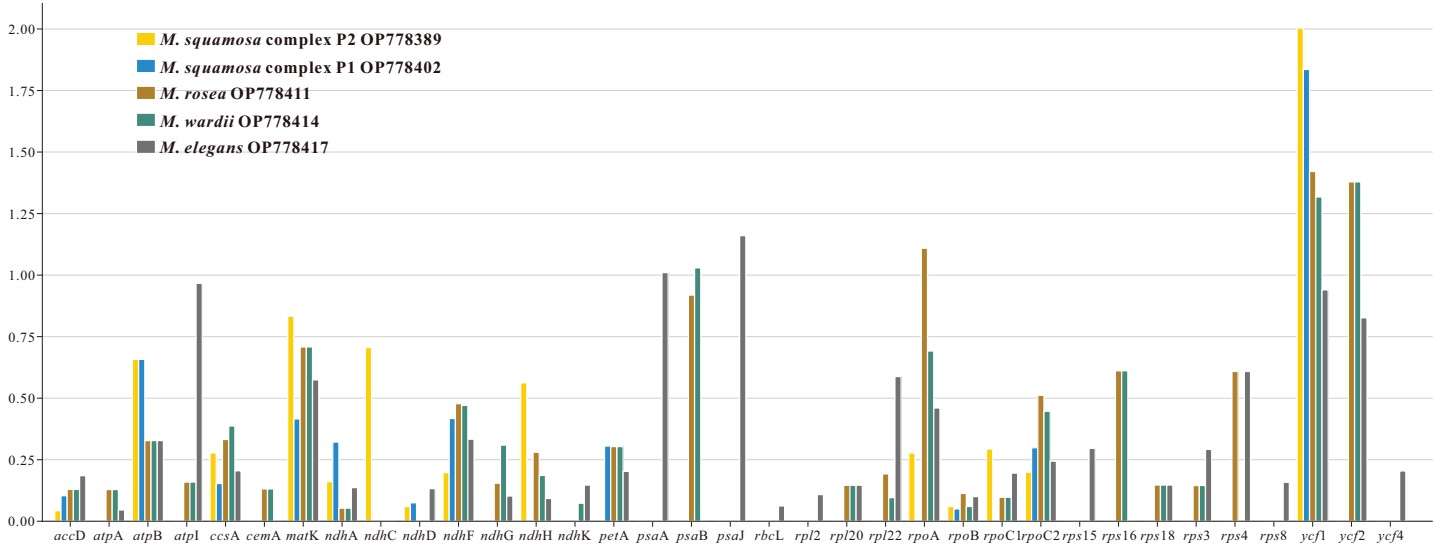

**Figure 3 The Ka/Ks values of 35 protein-coding genes (PCGs) of five *Myricaria* chloroplast genomes for comparison with *M. laxiflora*.** Ka, nonsynonymous; Ks, synonymous. A Ka/Ks ratio >1 indicates strong positive selection.

2,142–2,164 codons, while cysteine was the least abundant, with 223–228 codons (Table S3). In contrast, methionine and tryptophan were encoded by only one codon each, with codon counts ranging from 470 to 477 and 363 to 378, respectively, and neither showed codon usage bias (RSCU = 1). Among the six plastomes, the UUA codon for leucine had the highest RSCU values (2.00–2.04) and the AGC codon for Serine had the smallest RSCU values (0.33; Table S3). In general, codon usage and amino acid frequency were similar across the *Myricaria* taxa.

The nonsynonymous (Ka) and synonymous (Ks) substitution ratios were calculated for 80 shared CDSs in the six *Myricaria* plastomes, using *M. laxiflora* as the reference (Table S4). Out of these orthologous CDSs, 42 genes, including *clp*P, *inf*A. three *atp*, three *ndh*, five *pet*, two *psa*, 15 *psb*, six *rpl*, four *rps* and two *ycf* genes, had no nonsynonymous rate change. Additionally, 17 of these genes, as well as the *ndh*B and *rps*7 genes, had no synonymous rate change. Ka/Ks values of the remaining 35 CDSs ranged from 0.0430 (*acc*D. to 2.0038 (*ycf*1). Most of the genes had a Ka/Ks < 1, ranging from 0.0430 (*acc*D. to 0.9666 (*atp*I). However, six genes—*ycf*1 *ycf*2, *rpo*A, *psa*B, *psa*A and *psa*J—occasionally showed Ka/Ks >1 (Fig. 3).

### Sequence repetition in the Myricaria plastomes

A total of 373 SSRs were found in the six plastomes of *Myricaria*, and six types of repeat patterns were identified: mono-, di-, tri-, tetra-, and penta-nucleotide, as well as compound repeats. Mononucleotide repeats were the most abundant, accounting for 61.39% of the total SSRs ($n = 229$, ranging from 36 to 43), followed by dinucleotide (15.01%), trinucleotide (9.38%), tetranucleotide (7.51%), compound (6.17%) and pentanucleotide, which was extremely rare (0.54%; Table 2). Further comparison of the size and position of different SSR units revealed that composite SSR varied among the six samples, while
**Table 2 Type, location and number of simple sequence repeats (SSRs), tandem repeats and dispersed repeats found in the six *Myricaria* chloroplast genomes.**

| | Type | Numbers of repeats | | | | | | |
|---|---|---|---|---|---|---|---|---|
| | | *M. squamosa* complex P1 | *M. squamosa* complex P2 | *M. laxiflora* | *M. wardii* | *M. rosea* | *M. elegans* | Percentage (%) |
| SSRs | Mononucleotide | 43 | 36 | 37 | 37 | 40 | 36 | 61.39 |
| | Dinucleotide | 9 | 8 | 8 | 10 | 11 | 10 | 15.01 |
| | Trinucleotide | 6 | 6 | 7 | 6 | 5 | 5 | 9.38 |
| | Tetranucleotide | 5 | 4 | 6 | 4 | 4 | 5 | 7.51 |
| | Pentanucleotide | 0 | 0 | 0 | 1 | 1 | 0 | 0.54 |
| | Compound | 4 | 5 | 3 | 4 | 4 | 3 | 6.17 |
| | Total | 67 | 59 | 61 | 62 | 65 | 59 | — |
| Location | IGR | 40 | 39 | 40 | 44 | 45 | 38 | 65.95 |
| | CDS | 11 | 10 | 11 | 10 | 10 | 10 | 16.62 |
| | CDS-IGR | 1 | 0 | 1 | 0 | 0 | 0 | 0.54 |
| | Intron | 14 | 9 | 8 | 7 | 8 | 10 | 15.01 |
| | Exon | 1 | 1 | 1 | 1 | 2 | 1 | 1.88 |
| Tandem repeats | Total | 22 | 24 | 23 | 21 | 24 | 15 | — |
| Location | IGR | 14 | 15 | 14 | 11 | 13 | 8 | 58.14 |
| | CDS | 7 | 8 | 8 | 8 | 8 | 6 | 34.88 |
| | CDS-IGR | 1 | 1 | 1 | 1 | 1 | 1 | 4.65 |
| | Intron | 0 | 0 | 0 | 1 | 2 | 0 | 2.33 |
| Dispersed repeats | Forward | 15 | 16 | 13 | 13 | 17 | 12 | 36.29 |
| | Reverse | 0 | 0 | 9 | 2 | 4 | 9 | 10.13 |
| | Palindrome | 19 | 22 | 23 | 15 | 15 | 22 | 48.94 |
| | Complement | 0 | 0 | 4 | 0 | 1 | 6 | 4.64 |
| | Total | 34 | 38 | 49 | 30 | 37 | 49 | — |
| Location | IGR | 14 | 16 | 41 | 14 | 21 | 41 | 62.03 |
| | CDS | 9 | 9 | 0 | 1 | 2 | 0 | 8.86 |
| | CDS-IGR | 2 | 2 | 2 | 3 | 2 | 2 | 5.49 |
| | Intron | 5 | 7 | 5 | 8 | 8 | 5 | 16.03 |
| | tRNA | 3 | 3 | 1 | 3 | 2 | 1 | 5.49 |
| | Exon | 1 | 1 | 0 | 1 | 2 | 0 | 2.11 |

**Note:**
IGR, intergenic region.

mononucleotide repeats of A/T and dinucleotide repeats of AT/TA were almost all conserved (Table S5).

A total of 129 tandem repeats were found, ranging from 6 to 65 bp. The number of tandem repeats varied among the samples, with *M. elegans* having the fewest (15), and *M. squamosa* complex P2 and *M. rosea* having the most (24; Tables 2 and S6).

There were 237 dispersed repeats detected, belonging to four categories: forward, reverse, complementary and palindromic repeats. Palindrome repeats were the most
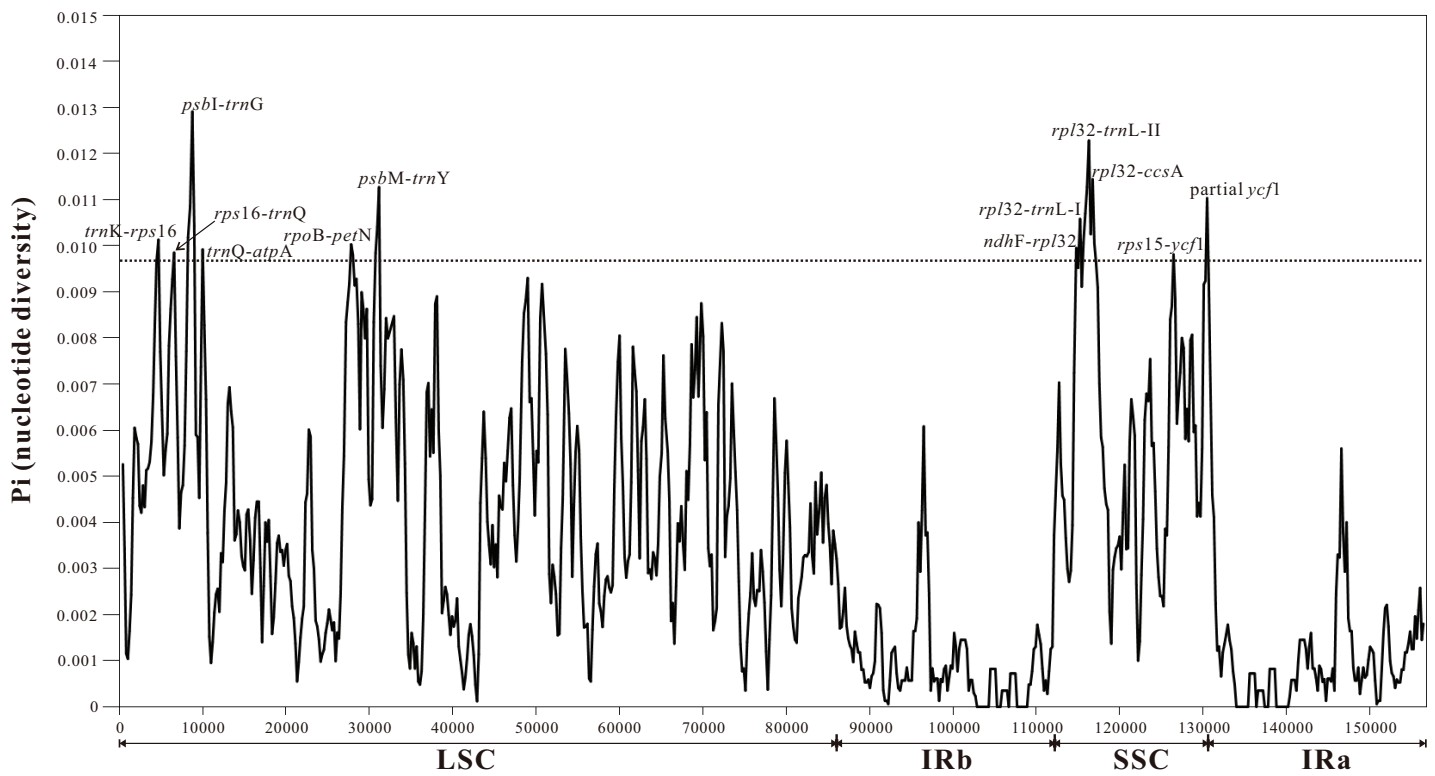

**Figure 4 Sliding window analysis of the entire plastome sequences of *Myricaria* species (window length: 800 bp; step size: 200 bp).** X-axis: position of the window, Y-axis: nucleotide diversity (Pi) of each window.

abundant (*n* = 15–23, 48.94%), followed by forward repeats (*n* = 12–17, 36.29%; Tables 2 and S7).

All individuals showed similar distributions of both type and location of SSRs, tandem repeats and dispersed repeats. The majority (58.14–65.95%) of these repetitive elements were located in the intergenic regions (IGRs; Table 2).

### *Myricaria-specific chloroplast barcodes*

*P*i values were used to determine hypervariable regions with potential to develop as group-specific barcodes for *Myricaria*. The sliding window analysis result showed that *P*i values in IRs were less than those in the LSC and SSC regions, ranging from 0 to 0.01291. A total of 11 highly-variable regions (*P*i > 0.0097, more than 75% of the maximum) were identified in *Myricaria* plastomes, including ten intergenic spacer regions (*trn*K$^{UUU}$-*rps*16, *rps*16-*trn*Q$^{UUG}$, *psb*I-*trn*G$^{UCC}$, *trn*Q$^{UUG}$-*atp*A, *rpo*B-*pet*N, *psb*M-*trn*Y$^{GUA}$, *ndh*F-*rpl*32, *rpl*32-*trn*L$^{UAG}$, *rps*32-*ccs*A, *rps*15-*ycf*1) and one protein-coding region (partial *ycf*1). These sequences were all located in LSC and SSC regions, with none found in IR regions (Fig. 4).

### *Complete plastome sequence comparison of Myricaria species*

Six *Myricaria* plastomes from different lineages in the plastome phylogenetic result were compared to visualize the overall sequence divergence (Fig. 5). The intergenic spacers located in single copy regions, namely *trn*E$^{UUC}$-*trn*T$^{GGU}$, *rps*4-*trn*T$^{UGU}$, *trn*F$^{GAA}$-*ndh*J, *pet*A-*psb*J and *ndh*F-*rpl*32, were found to be the most diverse regions.

Reference *Myricaria laxiflora* (OP778385): 1-154896 bp

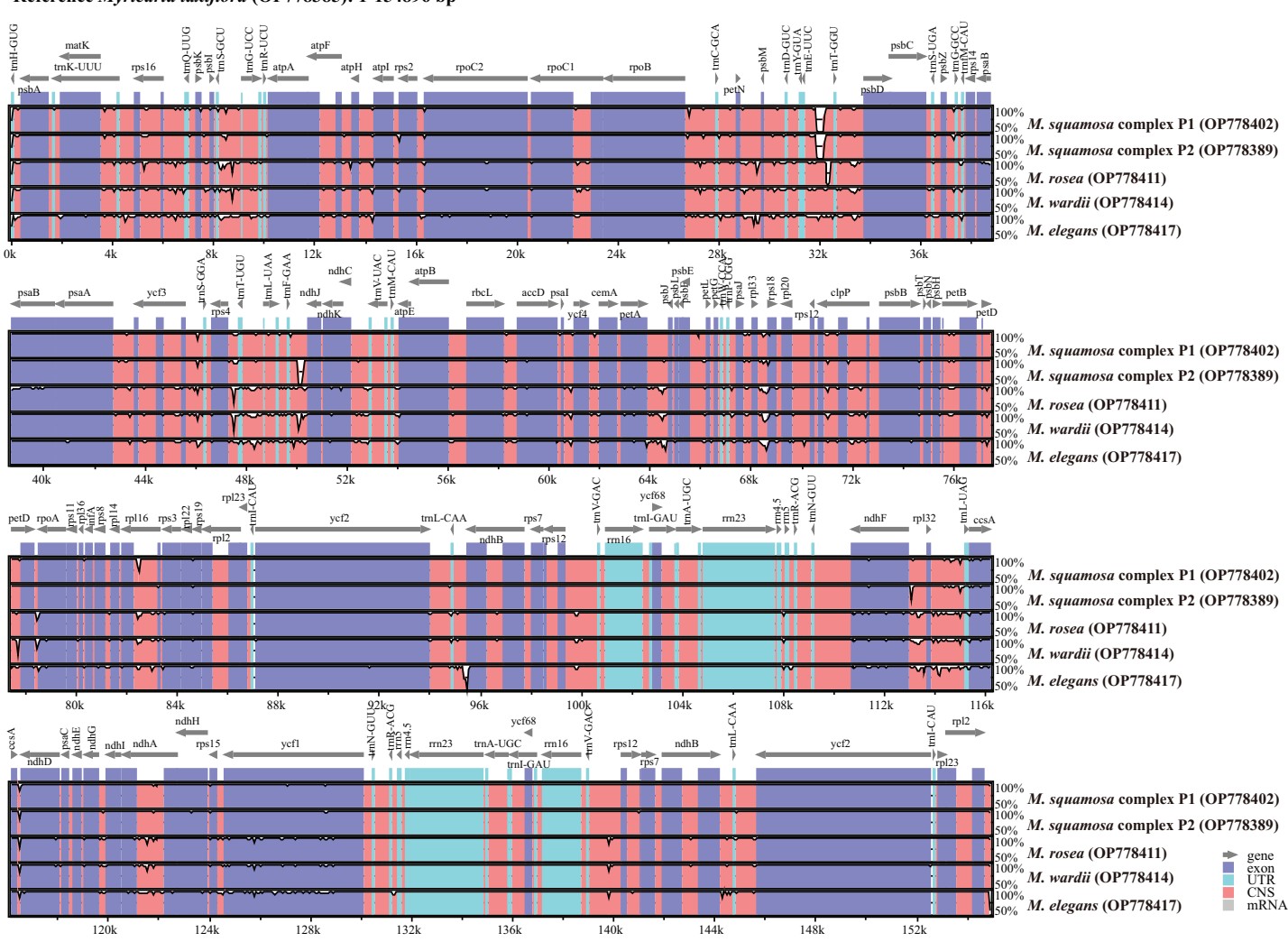

**Figure 5 Global alignment of chloroplast genomes of the six *Myricaria* samples using the annotated *M. laxiflora* (GenBank accession: OP778385) as the reference.** The x-axis represents the base sequence of the alignment, and the y-axis represents percent identity (50–100%). Annotated genes and their orientation are displayed using dark gray arrows. Genome regions are blue, purple, red and grey-coded as untranslated regions (UTR), exon, conserved non-coding sequences (CNS) and mRNA, respectively.

Comparing the IR/LSC and IR/SSC boundaries in the six chloroplast genomes uncovered stable IRs with little expansion or contraction (Fig. 6). The LSC-IRb borders were found to be located within the *rps*19 gene, with a shift of 38 bp in *M. elegans* and 80 bp in *M. laxiflora*. The LSC-IRa borders were located within the *trn*H$^{\text{GUG}}$ gene. The boundary of SSC-IRb was positioned one base in front of the *ndh*F gene, while SSC-IRa was positioned within the *ycf*1 gene.

## Whole plastome as super-barcode for phylogenetic reconstruction of *Myricaria* and species discrimination in the *M. squamosa* complex

The best-fit model of nucleotide substitutions for the ML and BI analysis was TVM+G, as calculated by jModeltest based on the matrix of the 43 whole plastome sequences aligned

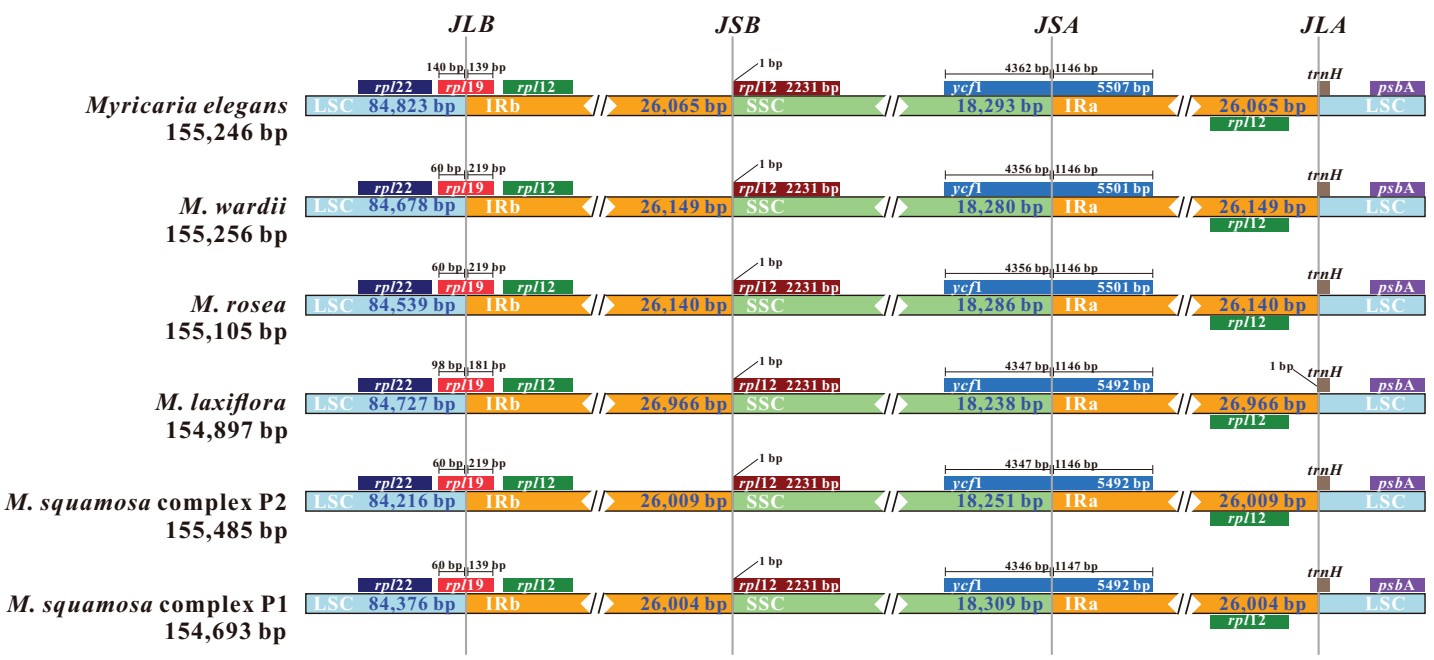

**Figure 6 Comparison and visualization of the boundaries of the IR, SSC and LSC regions among six *Myricaria* chloroplast genomes.** JLB: LSC/IRb; JSB: IRb/SSC; JSA: SSC/IRa; JLA: LSC/IRa. Numbers above the gene features denote the distance between the gene borders (either the start or end of the genes) and the junction sites.

by MAFFT (163,629 bp in length). The ML tree was generally congruent with the Bayesian consensus tree (Fig. 7A). The *Myricaria* samples formed six monophyletic lineages within two clearly separated clades, and all posterior probabilities (PP) and bootstrap supports (BS) were 100%. Clade A consisted of the well-delineated *M. elegans* and its variation (as lineage P6), and clade B grouped the remaining *Myricaria* species sampled in this study. The *M. squamosa* complex was separated into two lineages (P1 and P2), which were sister to *M. laxiflora* (P3), and together, were sister to a subclade consisting of *M. wardii* (P4) and *M. rosea* (P5). In total, the plastome sequences identified six well-supported monophyletic lineages. Four of these lineages corresponded to recognized species, namely *M. laxiflora*, *M. wardii*, *M. rosea* and *M. elegans*, and were successfully discriminated with high support values. However, *M. squamosa*, *M. bracteata* and *M. paniculata* were found to be non-monophyletic. Two newly-recovered lineages within cluster I, P1 and P2, were affiliated with three original taxa from the *M. squamosa* complex, but had no distinct morphological characteristics. Lineage P1 was mainly distributed in the eastern QTP, containing 21 populations originally assigned to three taxa of the *M. squamosa* complex, while P2 was distributed in the western Tarim Basin, consisting of four populations assigned to the three taxa of the *M. squamosa* complex (Fig. 7D).

Sequence divergences among 39 *Myricaria* plastomes were compared using nucleotide differences and sequence distances (Table S8). At the inter-species level, the greatest differentiation occurred between *M. laxiflora* and *M. elegans* ($p$-distance = 8.46 × 10⁻³),

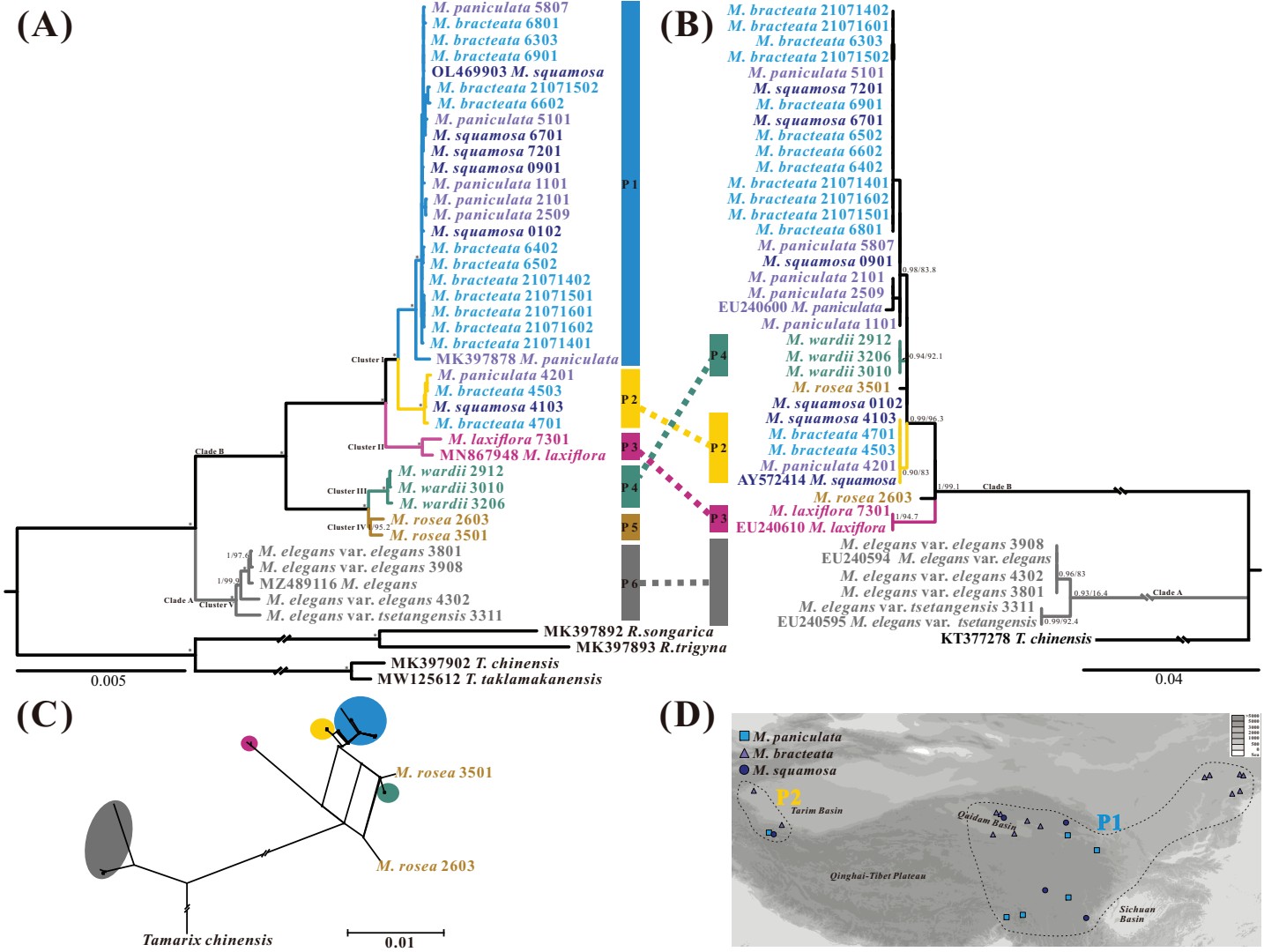

**Figure 7 Phylogenetic relationships of 39 *Myricaria* plastome accessions (A, left) and 40 ITS sequences (B, right) inferred from Maximum Likelihood (ML) and Bayesian Inference (BI) trees, a consensus network of the ITS and plastome ML trees generated using the Consensus Network method in SplitTree4 (C), and a distribution map of *M. squamosa* complex samples (D).** (A) The posterior probabilities (PP) and bootstrap supports (BS) are indicated above or below the branches. Branches with * have PP = 1 and BS = 100%. The colored column to the right of the tree indicates putative phylogenetic "species" delimited through plastome phylogenetic analyses (P1–P6). Of the 4–8 digits after the original species name, the last two digits represent individual numbers and the first 2–6 digits indicate population codes. It is suggested that lineages P1 and P2 should be integrated as a single species. The five tentative species suggested and their original taxonomic treatments are marked by different colors. (B) ITS dataset shows different inter-species relationships and poor species resolution. (C) The topology of the network shows slight incongruence between plastome and ITS phylogenetic trees. A distribution map (D) shows the distribution of 25 samples from the *M. squamosa* complex across different populations. P1 samples were mainly collected from the western Tarim Basin. Samples of P2 were collected from the northeast and southeast of the Qinghai-Tibet Plateau, as well as the Lüliang and Yin Mountains.

whereas the closest species relationship was found between *M. wardii* and *M. rosea* ($p$-distance = $1.01 \times 10^{-3}$). At the intra-species level, the maximum $p$-distances within species ranged from $2.13 \times 10^{-4}$ (*M. wardii*) to $1.38 \times 10^{-3}$ (*M. elegans*). The $p$-distances also supported the tentative taxonomic treatment of the five species using a tree-based method.

## Phylogenetic reconstruction and species discrimination of potential barcodes

The aligned nrITS matrix, consisting of 41 samples (35 assembled from genome skimming, five downloaded from GenBank, and one treated as an outgroup), had a length of 648 bp. The most appropriate model of nucleotide substitutions for the ML analysis was TIM2+G, as calculated by jModeltest. The phylogenetic analyses of the nrITS sequences revealed two distinct parallel clades, similar to the plastome dataset: the poorly-supported *M. elegans* clade (clade A, BS = 16.4%, PP = 0.93), and the well-delineated clade B, comprising the remaining *Myricaria* species. However, the subclades recovered within the *Myricaria* species clade (clade B) and their relationships differed from those observed in the plastome dataset. Notably, only *M. laxiflora*, *M. wardii* and P2 clustered separately, whereas *M. rosea* accessions did not group together (Fig. 7B). A few individuals from P1 of the *M. squamosa* complex (codes 0901 and 0102) failed to form a monophyly. *M. wardii* appeared to be totally embedded among *M. squamosa* complex accessions. Overall, the nrITS dataset identified only three monophyletic lineages, two of which corresponded to recognized species, *M. laxiflora* and *M. wardii*. A phylogenetic network, reconstructed using the ConsensusNetwork method (Fig. 7C), unveiled the presence of two well-defined clades, *M. elegans*, and the rest of the *Myricaria* species, which had apparent clustering patterns. While the topology of the network exhibited slight incongruence with the phylogenetic trees constructed using the plastome and ITS datasets (Fig. 7), the species group delineated by the phylogenetic network closely resembled those trees, except for *M. rosea*. Within the *M. squamosa* complex, there were two lineages (P1 and P2) with alternative splits connecting different sections of *M. laxiflora*, *M. wardii* and *M. rosea*. This reticulate network rendered the phylogenetic placement of these closely related species unclear.

The aligned standard chloroplast DNA barcode combination matrix was 4,417 bp in length. The ML analysis of the four-barcode combination revealed four monophyletic lineages (Fig. 8A). Similar to the phylogenetic analyses of the plastome and ITS datasets, the *Myricaria* samples formed two clades; the *M. elegans* clade, which was highly supported in this dataset, and the remaining *Myricaria* species. The *M. squamosa* complex clustered together and was sister to *M. laxiflora*, while *M. rosea* individuals were paralleled with *M. wardii*. In total, the standard chloroplast DNA barcodes successfully identified four monophyletic lineages corresponding to the *M. squamosa* complex, *M. laxiflora*, *M. wardii* and *M. elegans*.

Further phylogenetic analyses were performed using each region and a combination of newly-obtained *Myricaria*-specific barcodes. The relationships between each monophyletic lineage were generally consistent with the plastome dataset, although the relationship between the *M. squamosa* complex and *M. laxiflora* showed slight differences. The single *Myricaria*-specific barcode and 11-barcode combination both confirmed the presence of up to five monophyletic lineages corresponding to four recognized species (Figs. 8B and S1).

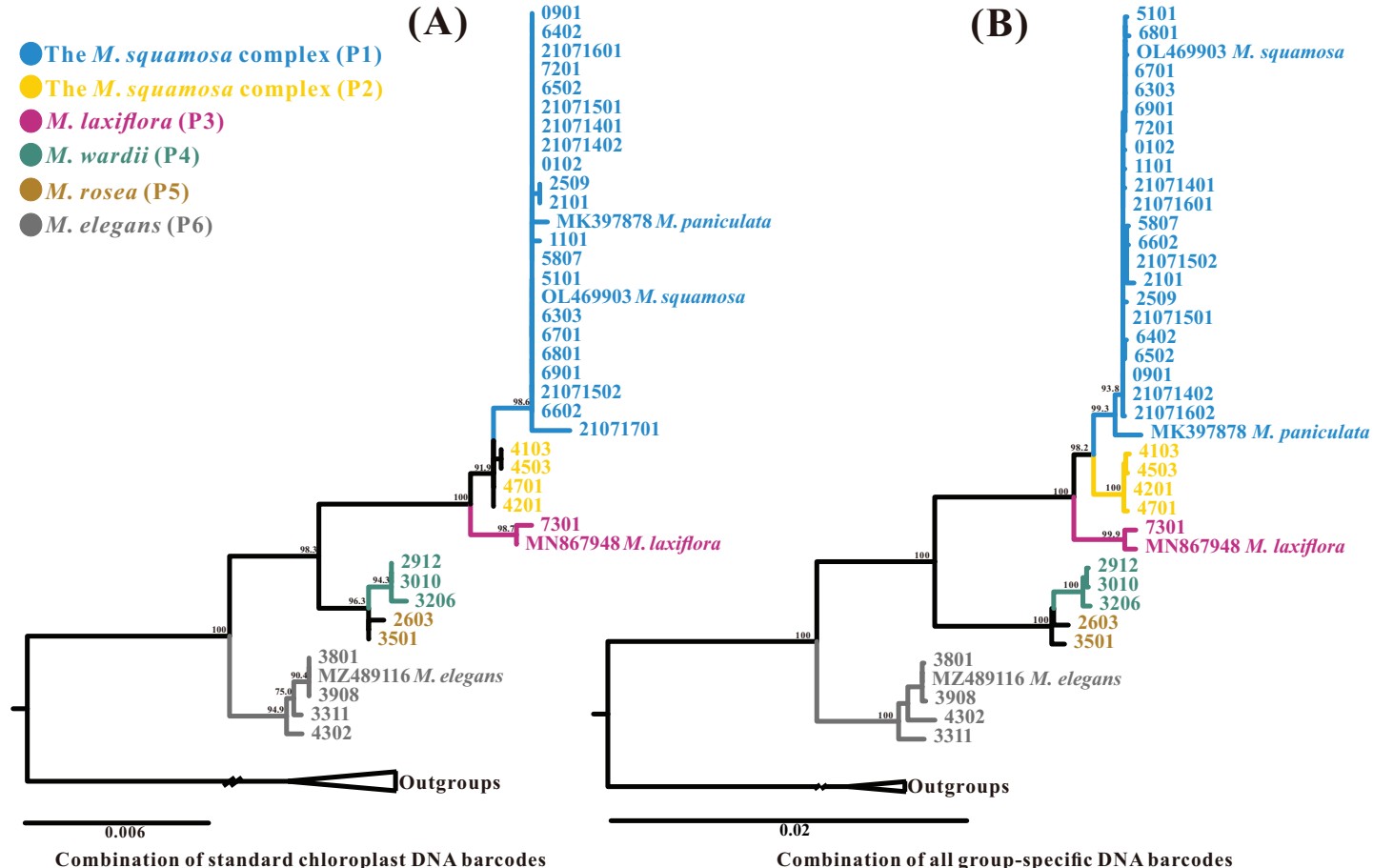

The M. squamosa complex (P1)
The M. squamosa complex (P2)
M. laxiflora (P3)
M. wardii (P4)
M. rosea (P5)
M. elegans (P6)

(A)

Combination of standard chloroplast DNA barcodes

(B)

Combination of all group-specific DNA barcodes

**Figure 8** Phylogenetic relationships generated through ML analysis of the standard plastid DNA barcode combination (A, *mat*K+*rbc*L+*trn*H-*psb*A+*trn*L-F) and a group-specific DNA barcode combination (B, consisting of 11 highly-variable regions–*trn*K-*rps*16+*rps*16-*trn*Q+*psb*I-*trn*G +*trn*Q-*atp*A+*rpo*B-*pet*N+*psb*M-*trn*Y+*ndh*F-*rpl*32+*rpl*32-*trn*L+*rps*32-*ccs*A+*rps*15-*ycf*1+partial *ycf*1). The posterior probabilities (PP) are indicated above the branches. The terminal 4–8 digits on the cladogram represent sample codes. The putative phylogenetic "species" delimited through plastome phylogenetic analyses (P1–P6) are marked by different colors.               

## DISCUSSION

### *Myricaria* chloroplast genome features and genome variations

This work conducted the first-ever comparative analysis of *Myricaria* chloroplast genomes. The results showed that all *Myricaria* chloroplast genomes had a typical quadripartite structure containing two IR regions, each separated by a LSC and SSC region (Fig. 2), as reported for other land plants (*Wicke et al., 2011*; *Shi et al., 2023*). The *Myricaria* chloroplast genomes ranged from 154,485 to 155,347 bp in length and displayed high conservation, with only minor differences mainly caused by expansion or contraction of the IR regions (Fig. 6; *Yang et al., 2023*). Significant IR contractions of *rps*19 and *ycf*1 were observed, which were predicted as the main contributors to the overall variation observed among *Myricaria* chloroplast genomes, in line with reports for species in other genera, including *Cerasus*, *Prunus* and *Rubus* (*Yu et al., 2022*; *Wan et al., 2023*). The comparative analysis and *P*i values test (Figs. 4 and 5) showed that the non-coding regions of the LSC

and SSC regions exhibited higher divergence compared to the IR regions, which is consistent with the findings of similar studies (*Yu et al., 2022*; *Yang et al., 2023*). Additionally, 11 *Myricaria*-specific barcodes were identified that have the potential of reflecting inter-species relationships, with 10 positioned in the IGRs of the LSC and SSC regions (Figs. 4, 8B and S1).

A total of 114–115 unique genes were annotated in this study (Tables 1 and S1), including four rRNA genes, 30 tRNA genes, and 80–81 CDSs, similar to previous reports (*Liu et al., 2020b*). There were some differences in the genes annotated in this study compared to previous studies, mainly reflected in CDSs, such as *ycf* genes (*Wan et al., 2023*); specifically, the *ycf*15 gene was detected in 32 *Myricaria* samples, but absent in three individuals (codes 7301, 3010 and 2912; Table S1). The discrepancy in the number of PCGs in the publicly-available *Myricaria* plastomes may be attributed to a different annotation strategy.

## Evolutionary and phylogenetic analysis of the genus *Myricaria*

Repetitive sequences are crucial in the rearrangement and diversification of chloroplast genomes, making them essential for studying indels and substitutions (*Cavalier-Smith, 2002*; *Shi et al., 2023*). A total of 129 TRs and 237 dispersed repeats were discovered in this study, with the majority of them located in the IGRs of the LSC region (Tables 2, S6 and S7), which is similar to the findings of previous studies (*Ruang-Areerate et al., 2021*). Chloroplast SSRs are highly polymorphic and valuable markers for identifying population genetic structure and phylogeography patterns at both inter- and intra-population levels. A total of 337 SSRs were detected in this study, with six types in the *Myricaria* chloroplast genomes and a predominant mononucleotide repeat of A/T (Tables 2 and S5).

Codons serve as a crucial foundational element linking amino acids, proteins, and genetic materials in living organisms (*Wang et al., 2022b*). RSCU results indicated that *Myricaria* chloroplast genes tend to end with A/T codons, which is consistent with observations made in other plants (*Campbell & Gowri, 1990*). The *Myricaria* plastomes exhibited remarkable similarity in GC1, GC2, GC3, GC3s, codon usage, and amino acid frequency, which is likely related to the conservation of chloroplast genomes within this taxonomic group.

Ka and Ks nucleotide substitution rates as well as the Ka/Ks ratio are commonly used to estimate the differences in gene sequences and potential purifying selection in CDSs. A selective pressures analysis was performed on the orthologous CDSs, and the results showed that the Ka/Ks ratio for most genes was less than 1, supporting the presence of purifying selection in the CDSs of *Myricaria* chloroplast genomes (*Makalowski & Boguski, 1998*). However, there were six genes that showed evidence of strong positive selection, suggesting potential functional divergence or adaptive evolution.

Until now, phylogenetic analyses of the genus *Myricaria* have been based only on ITS and cpDNA markers, which have limited variations (*Liu, Wang & Huang, 2009*; *Zhang et al., 2014*). In this study, the inter-lineage relationships identified by the maternally-inherited plastomes differed from those identified by the biparentally-inherited

nuclear ITS sequence variations (Figs. 7A and 7B). The position of *M. elegans* has been a matter of debate, with arguments for its placement within *Myricaria*, *Tamarix*, or even as a new genus called *Myrtama*, an intermediate and hybrid group between *Myricaria* and *Tamarix* (*Baum, 1966*; *Zhang et al., 2003*; *Gaskin et al., 2004*; *Hua, Zhang & Pan, 2004*; *Zhang et al., 2014*; *Channa, Shinwari & Ali, 2018*). The plastome results of the present study (Fig. 7A) further support the inclusion of *M. elegans* within the genus *Myricaria*, as all of the *Myricaria* species, including *M. elegans*, showed strong support with *Myricaria*, with a bootstrap support (BS) of 100% and a posterior probability (PP) of 1. While the ITS tree in this study (Fig. 7B) showed a parallel relationship between *M. elegans* and the remaining *Myricaria* species, it did not provide sufficient evidence for including *M. elegans* within *Myricaria* due to insufficient variability features. The *M. squamosa* complex was sister to *M. laxiflora* on the plastome tree, but had a closer relationship with *M. wardii* on the ITS tree. Notably, none of the interspecies relationships inferred from the plastome variations were confirmed by the ITS dataset. Phylogenetic inconsistencies between different genes or genomes, particularly between plastome and ITS datasets, are prevalent in most angiosperms (*Rokas & Chatzimanolis, 2008*; *Hu et al., 2016*; *Villar et al., 2019*; *Giaretta et al., 2021*; *Su et al., 2021*). Biological and methodological factors, such as incomplete lineage sorting (ILS), hybridization, introgression in species undergoing rapid radiation and convergent molecular evolution, as well as sample error, rate signal, model selection and heterotachy, can contribute to conflicting gene trees (*Zhang et al., 2020*; *Cai et al., 2021*; *Doyle, 2021*; *Steenwyk et al., 2023*). The occurrence of hybridization processes in Tamaricaceae has frequently been reported, even between extremely different taxa (*Gaskin & Schaal, 2002*; *Mayonde et al., 2015*). For example, reticulate gene flow might have occurred between *M. wardii* and *M. rosea* and the *M. squamosa* complex. It is also highly probable that ILS in *Myricaria* contributes to the persistence of ancestral genetic polymorphisms during fast speciation events, resulting in phylogenetic discordance between plastome and ITS datasets. Wider taxonomic sampling and nuclear variations with sufficient information are necessary to better understand this significant heterogeneity and to further explore possible causes of these incongruences (*Gonçalves et al., 2019*).

## Species boundaries within the *M. squamosa* complex

Results of the present study on the tree-based species boundaries of seven recognized *Myricaria* species in China revealed two broad findings:

1) Despite the difficulties in delimiting species within the *M. squamosa* complex based on variable flower morphology in a field investigation (Fig. 1), phylogenetic analyses of plastid datasets, which accumulated mutations, highly supported the incorporation of three traditionally recognized taxa, *M. squamosa*, *M. bracteata* and *M. paniculata*, into one species (Figs. 7A, 8 and S1). This suggests the presence of two paraphyletic, independent ESUs (P1 and P2) within this species complex, but indicates both are still monophyletic and likely undergoing speciation.

2) Complete chloroplast genome sequences provide higher species resolution compared to much shorter standard DNA barcode fragments (two datasets: ITS, and four plastid barcode combination), allowing for the distinction of six well-supported monophyletic lineages. For P1 and P2, which may be revised as one taxon, the entire chloroplast genome sequences were able to discriminate all five tentative species. In contrast, the ITS region resolved only three lineages, representing two species (Fig. 7B), and standard plastid DNA barcodes resolved four lineages, representing four species (Fig. 8A). To further reduce costs, 11 group-specific barcodes were selected from plastomes based on high *P*i values. Among them, a combination of barcodes and four single barcodes showed the highest discrimination rate of four out of five species, identifying five lineages (Figs. 8B and S1). Though all barcodes failed to detect the *M. rosea* lineage, the *trn*K-*rps*16, *rps*16-*trn*Q, *psb*M-*trn*Y and *rps*15-*ycf*1 regions are still suitable candidate group-specific barcodes that offer different details in species delimitation. These findings contribute to an improved understanding of the widely-used complete plastid genome as a super-barcode for distinguishing closely related species (*Wang et al., 2018*; *Dong et al., 2021*).

The results of this study support the adoption of five *Myricaria* clusters as species (Fig. 7). Phylogenetic results showed that all six lineages within the five clusters formed distinct evolutionary units, satisfying the criteria for phylogenetic species (*de Queiroz & Donoghue, 1988*). However, dividing the *M. squamosa* complex into two distinct taxa based on the newly-recovered reciprocally monophyletic lineages (P1 and P2) would conflict with the former morphological criteria. Phenotypic gaps between species with genetic distinctions are often considered initial and critical evidence in different species delimitation methods (*Liu, 2016*). The five clusters have previously been described as seven independent species in the morphological species concept (*Zhang & Zhang, 1984*; *Yang & Gaskin, 2007*). However, determining species boundaries between *M. bracteata*, *M. paniculata* and *M. squamosa* is challenging due to the lack of clear morphological divergence. The morphological gaps previously utilized to delimit these three taxa appear to contradict the phylogenetic results. No morphological divergence was observed after further subdivision of Cluster I into two ESUs, P1 and P2. The P2 lineage, comprising four populations in the western Tarim Basin (Populations 41, 45, 42, 47), exhibits three distinct morphological characteristics. Population 41, characterized by lateral racemes on old branches, and solitary or clustered racemes in axils, has been classified as *M. squamosa* based on field surveys and specimen records. Population 42, distinguished by a typical large panicle terminal on current-year branches starting in September, has been classified as *M. paniculata*. Racemes of populations 45 and 47 were terminal on current-year branches, clustered into spikes, and their bracts were about 1.5 times broader than those of populations 41 and 42, supporting their classification as *M. bracteata*. Samples of P1, collected from the eastern QTP, previously belonged to three different taxa based on minor morphological differences in inflorescence type, bract size and imbricate scales, but now form a well-supported monophyletic evolutionary lineage. The presence of interspecific morphological differences in both P1 and P2 ESUs was confirmed based on sufficient field

surveys and specimen records. However, such differences lack discontinuous, nonoverlapping morphological traits, which indicates that they cannot be diagnosed solely based on the criteria of the morphological species concept. Considering genetic distinctions, phenotypic gaps and distribution evidence, *M. bracteata* and *M. paniculata* should be classified as two varieties under *M. squamosa*, as phenotypic plasticity may result from adaptation to different environments (*Stern, 2013*; *Cheng et al., 2018*). Rapid environmental changes and habitat fragmentation caused by anthropogenic activities can lead to swift phenotypic changes in species, which may not necessarily reflect evolutionary responses over deep timescales (*Levis & Pfennig, 2016*). For instance, the Ericales displayed elevated conflict and rapid phenotypic change during early radiation (*Larson et al., 2020*). The rates of morphological evolution vary significantly across plants, and the underlying causes of these patterns remain unclear but merit further study. The observed discordance between genetic and morphological divergence in the *M. squamosa* species complex can be attributed to the complex interplay between the micro- and macroevolutionary processes that drive major organismal changes from the genomic to the phenotypic level (*Parins-Fukuchi, Stull & Smith, 2021*). It is also worth noting that morphological features do not always align with phylogenetic clades, even in closely related groups, as seen in *M. paniculata*, which typically has two different inflorescence types, but they do not group together. Similar difficulties have been encountered in the phylogenetic revision of *Tamarix*, such as the incongruence between morphological features and phylogenetic relationships of *T. amplexicaulis*, *T. canariensis* and *T. gallica* (*Villar et al., 2019*). The incongruence between molecular phylogeny and morphological classification reported here may also largely be attributed to ILS, where the uniparental attribute of plastid and the biparental-inherited ITS regions do not offer sufficient information to resolve the phylogeny (*Villar et al., 2019*; *Feng et al., 2022*; *Liu et al., 2022*).

## CONCLUSIONS

This study performed comparative analyses of plastomes for multiple *Myricaria* species at the population level and tested species resolution of whole plastomes, standard DNA barcodes and group-specific plastid barcode candidates in the *Myricaria*. The tree-based evidence from plastid data highlighted inconsistencies between molecular phylogenetics and traditional taxonomic systems, suggesting that *M. squamosa*, *M. bracteata* and *M. paniculata* should be treated as a single taxon. Moreover, this study provides valuable, comprehensive molecular markers for further species identification for all *Myricaria* taxa. Further field investigations of morphological distinctions among all *Myricaria* species are proposed, encouraging international cooperation with dense sampling and population genetics studies.

## ACKNOWLEDGEMENTS

The authors thank Mei Zhang and Linrui Yi for their technical assistance.

### Funding

This study was supported by grants from the National Natural Science Foundation of China (31860104), the Guizhou Science and Technology Program (QKHJC-ZK[2022]-591), Science and Technology Project of Zunyi City (ZSKHHZ[2020]88), and the Outstanding Young Talent Project of Zunyi Medical University (17zy-002). The funders had no role in study design, data collection and analysis, decision to publish, or preparation of the manuscript.

### Grant Disclosures

The following grant information was disclosed by the authors:
National Natural Science Foundation of China: 31860104.
Guizhou Science and Technology Program: QKHJC-ZK[2022]-591.
Science and Technology Project of Zunyi City: ZSKHHZ[2020]88.
Outstanding Young Talent Project of Zunyi Medical University: 17zy-002.

### Competing Interests

The authors declare that they have no competing interests.

### Author Contributions

- Huan Hu conceived and designed the experiments, performed the experiments, analyzed the data, prepared figures and/or tables, authored or reviewed drafts of the article, and approved the final draft.
- Qian Wang performed the experiments, authored or reviewed drafts of the article, and approved the final draft.
- Guoqian Hao performed the experiments, authored or reviewed drafts of the article, and approved the final draft.
- Ruitao Zhou analyzed the data, prepared figures and/or tables, and approved the final draft.
- Dousheng Luo analyzed the data, prepared figures and/or tables, and approved the final draft.
- Kejun Cao analyzed the data, prepared figures and/or tables, and approved the final draft.
- Zhimeng Yan performed the experiments, prepared figures and/or tables, and approved the final draft.
- Xinyu Wang conceived and designed the experiments, analyzed the data, authored or reviewed drafts of the article, and approved the final draft.

### Field Study Permissions

The following information was supplied relating to field study approvals (*i.e.*, approving body and any reference numbers):

Field experiments were approved by the Research Council of Zunyi Medical University (Project Number: 31860104, Identification Code: ZMU-BO-1903-169).

## DNA Deposition

The following information was supplied regarding the deposition of DNA sequences:

The annotated plastid genomes and ITS sequences datasets are available at NCBI GenBank: OP763799–OP763833, OP778385–OP778418, and OP756567.

## Data Availability

The raw data of NGS supporting the results are available at GenBank: SRR22582227–SRR22582261; PRJNA910234.

## Supplemental Information

Supplemental information for this article can be found online at http://dx.doi.org/10.7717/peerj.16642#supplemental-information.

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
