# Peer review of "Insights into the phylogenetic relationships and species boundaries of the Myricaria squamosa complex (Tamaricaceae) based on the complete chloroplast genome"

_PeerJ, doi:10.7717/peerj.16642_

## Round 0.1 · original submission · Major Revisions

i totally agree with Reviewer 1, the paper needs to be reorganized in all sections, from Hypothesis and Questions, Methodology, and Discussion, as indicated. In methodology explain clearly how the phylogenetic analyses were conducted, step by step with MrBayes. Please consider carefully every issue raised by the two reviewers, which coincided in that the English of the article does not reach a professional level, you can use an editorial service and state clearly that the paper was corrected by a fluent English speaker or a company.

**Language Note:** The Academic Editor has identified that the English language must be improved. PeerJ can provide language editing services - please contact us at copyediting@peerj.com for pricing (be sure to provide your manuscript number and title). Alternatively, you should make your own arrangements to improve the language quality and provide details in your response letter. – PeerJ Staff

Reviewer 1 ·

Excellent Review

This review has been rated excellent by staff (in the top 15% of reviews)
EDITOR COMMENT
The reviewer carefully and thoroughly reviewed every aspect of the manuscript, fom hypothesis, methods, results and discussion. Comments were clearly stated and very constructive. In my opinion if authors take into account every issue raised, the paper will improve greately. Thank you for your effort! one of the best reviews I have seen in the manuscripts submitted to PeerJ.

Basic reporting

The article “Insights into the phylogenetic relationships and species boundaries in the Myricaria squamosa complex (Tamaricaceae) based on the complete chloroplast genomes” provides new data about the phylogenetic history of the genus Myricaria. The article includes 35 new plastome sequences from seven Myricaria species. However, before considering the article for publication, it should be thoroughly revised.
One of the main issues with the article is that the order in which the methodology and results are shown is a bit odd. In my opinion, the article is clearly divided in two parts: the description of the characteristics of the plastid DNA in the genus, and the phylogenetic reconstruction of the species complex. The first half of the article is highly descriptive, but it can be of interest to future researchers that want to reuse the newly sequenced plastomes. However, I consider that the most interesting results can be found in the phylogenetic section. Considering this, I think that the article would improve if its layout were restructured. In the methodology and the results, the phylogenetic analyses (lines 248-304) are in the middle of the description of the different characteristics of the plastome. The phylogenetic and barcode analyses should be presented both in methodology and results after the more descriptive analyses of the plastome. In fact, the discussion section is already divided in two sections that correspond to these two parts that I have described. Besides, more importance should be given to the phylogenetic results in the discussion (see latter comments about this aspect).
The objectives of the article should also be rephrased. The second and third objectives are almost identical and could be combined into a single objective. Besides, the order of the objectives should roughly match the order in which the results are presented.
The English language should be improved to ensure that an international audience can clearly understand the text. There are numerous orthographical and typographical errors such as beginning a sentence with M. (l.40), Siberaia (line 45), a space after plastomes (l. 147), squasoma (l. 256), thel (l.293), suppoerted (l.294), Myriacia (l.300), capital letter in complete (l.305), Myricaria (l.360), Myrcaria (caption of Figure 2), among many others. It is important to pay a special attention to the name of the species [it is M. paniculata, not M. paniculate (l.4, 44)], or to the standard name of the authors following IPNI [Linneo is abbreviated as L., not Linn. (l.27,30)]. Some other examples where the language could be improved include lines 37, 41-42, 47-49, 53-54, 163, 234-235, 255, among others– the current phrasing makes comprehension difficult. I suggest you have a colleague who is proficient in English and familiar with the subject matter review your manuscript, or contact a professional editing service.
Raw sequencing data should be deposited into a public database, not only the ITS and plastome. It should be clear in the text whether it has been done or not, and how to access the data.
I consider that the figures are appropriate in general, although the size of the text could be increased in some figures to increase legibility, such as Figures 3 and 5). However, Figure 4A needs extensive changes. First of all, the figure would be much more legible if the tree was rotated 90º to the left. If done, the name of the species could be read from left to right. The size of the text should be increased and the color chosen for some text could be changed to improve legibility. Besides, I think that a cladogram is not the best type of tree given the purpose of the article. The fact that all nodes are separated by the same distance usually gives the false impression that groups are much more different that they actually are. Consider changing the tree by a phylogram, since the genetic distance between the different clusters can be perceived visually.

Experimental design

The experimental design is correct and appropriate to the proposed objectives. However, some aspects should be clarified.
I have some questions about the analysis with MrBayes. The most common analysis with this program uses four independent chains that have different temperatures that help to better explore the parameters, which is called the Metropolis coupled Markov Chain Monte Carlo (MC3) algorithm (a variant of the MCMC that you say you performed). Please check if this is the case and if so, correct the text, since the MC3 method is the predetermined method in MrBayes. Besides, the sampling was performed every 100 generations, which seems a bit too often for me. Sampling too often can create problems if the samples are correlated. I recommend checking the correlation of the samples of the MCMC. This can be easily done with the free software Tracer (https://beast.community/tracer) by checking if the Estimated Sample Size (ESS) of all estimated parameters have at least a value of 200. If this value is bigger than 200 for all parameters, the analyses is correct and no changes need to be done. However, if ESS is lower than 200 for some parameters, you should repeat the MCMC sampling with a lower sampling frequency (for example, every 1000 generations) and more generations in total.
Please, also cite properly the Organellar Genome Draw (line 142), and explain the meaning of the parameters in line 208.

Validity of the findings

The results of the article provide interesting new data about the phylogenetic history of the genus Myricaria. However, I think that some conclusions are not completely supported by the results and some supplementary analyses could help to better understand the phylogenetic history of the genus.
The taxonomic status of the three species included in M. squamosa complex are revised as a single taxon in the article (lines 267-270), and two species are suggested to be varieties under M. squamosa (line 454). The main problem is that these taxonomic conclusions are drawn almost only from plastid data. Plastid genome is haploid in plants, and it is inherited exclusively in angiosperms from the seed and not from pollen. As a consequence, recombination between different plastid genomes is almost impossible and the whole chromosome is inherited as a whole, it behaves like a really big gene in terms of evolution. Other nuclear genes can have experienced different evolutionary processes, and hence considering that the gene tree of the plastid is necessarily the phylogenetic history of the species can lead to big mistakes. The article already includes data about a nuclear region, the ITS region, but its tree is in Supplementary material. Even though the tree is not well resolved, it gives valuable information about another gene tree in the genus. As a consequence, the ITS tree should be included in the main text. I should point out that I also think that probably the M. squamosa complex should be treated as a single taxon. However, I think that the analyses currently included in the text do not rule out other possibilities.
The plastid data of the M. squamosa complex clearly show a geographic pattern, differentiating west from east populations. Although the possibility of the whole complex being a single species is plausible, other processes can cause this pattern, like hybridization processes, or incomplete lineage sorting. This latter process is especially important in groups of species that have rapidly diverged and that had ancient population variability. For example, in the genus Tamarix (from the same family Tamaricaceae), Villar et al (2019) found strong discrepancies between nuclear and plastid trees. For example, for T. amplexicaulis, morphologically indistinguishable individuals that lived together in the same population can have extremely different plastid sequences. These processes are named in the discussion, but little importance is given to them. I think that these aspects should be expanded.
Villar, J. L., Alonso, M. Á., Juan, A., Gaskin, J. F., & Crespo, M. B. (2019). Out of the Middle East: New phylogenetic insights in the genus Tamarix (Tamaricaceae). Journal of Systematics and Evolution, 57(5), 488-507.
Besides, there are programs like SplitsTree that include several analyses to check for discrepancies between different gene trees. This program is fast and easy to use, and I consider that an analysis such as NeighbourNet or a consensus network of both trees (plastid and ITS) could help to check if there are observable discrepancies between both trees.
In relation to these aspects, are you sure about the morphological identification of the species? Is there any sign of morphological introgression between individuals that could suggest existing hybridization processes?
In line 476, it is stated that Myricaria is wind pollinated. As far as I know, the seeds are wind dispersed, but the flowers are pollinated by insects. Please check these aspects and the references included in those lines.
There are 23 Tables in Supplementary Material. Many of them have the exact same layout since they simply refer to different species, and hence could be joined into a single Table by including an additional column indicating the species. Tables S3 to S8 could be joined, as well as S9 to S14, and S15 to S20.
Other minor comments:
What do you mean by Ecological values (line 5)?
You say that Myricaria is on the verge of extinction (l.22), but no data or references are provided about this statement in the main text.
Please, cite Desvaux properly in line 27, you are referring to a specific reference from this author.
Abbreviatures should be explained in the main text, such as QTP in line 28.
In line 56 you say that there has been debate. You should include the main references about this debate so readers can follow it.
Citation of CBOL Plant Working Group (2009) is incorrect. Please check the proper way of citing this article (line 76).
Delete lines 217-219, they are not results, it has been previously stablished in previous sections.
Lines 232-239 should be in one paragraph, not two paragraphs.
The results section should not have references in it (line 261).
Lines 267-270 is discussion, not results. Please, change it to its section.

Additional comments

The article shows interesting new data about the phylogenetic history of the genus Myricaria, and I think that after a thorough revision it could be a good addition to the journal. Consequently, I think it should be considered for publication after major revisions.

Reviewer 2 ·

Basic reporting

English language is used to describe the article however at few places certain words can be rephrased to ensure clear understanding of your text. For example, in the abstract line number 13, the word “couldn’t” can be replaced with “could not be”. Similarly, in introduction line number 54 “do not” could not be used for branches, the whole sentence can be rephrased. Line number 87 in introduction “it hinted to us” does not seem to be appropriate. Line number 107 “so far failed” can be replaced by “traditional barcoding approach is insufficient”. I am not sure if using the word “we” is suitable in writing an article as it is seen used in lines 104, 108, 146 and so on.
Add space before in text reference at line number 147.
Occupying instead of occupied at line number 160.
Representing instead of represented at line number 179.
Comparison and rest of the analysis or comparison and further analysis at line number 180.
Replace which, use with instead in line number 240.
Line 255 remaining instead of remain
Line 256 should be “clustered separately into two lineages”
Line 280, 281 “and 1 accession was treated as outgroup, being 648bp long”.
“Myricaria s.str”. in lines 284.285 what is full form of s.str it is sister?
Line 287, the word “didn’t” group together followed by line 288, form a monophyly as plastomes “did”can be replaced with appropriate words with same meaning. Remove 1 from the in line 293. Correct spelling of supported in line 294.
Correct the spelling of “supporting” in line number 396.
Lines 417, 418, species not specie, “failed to” detect instead of “failed in”.
Line 437 “we didn’t find any morphological divergence” can be rephrased like “any morphological divergence could not be found in congruence”.
Line 440 and beyond, Pop used to abbreviate population although it has not been mentioned earlier it is better to use the full form.
Line 470, “in light of numerous studies have reported” can be replaced by “studies reporting”

Your introduction can be improved by rearranging certain paragraphs for consistency in the information provided. I suggest you could elaborate the need to focus on this particular complex in the first paragraph lines 36-39. In fact, you can add lines 84-89 in the first paragraph for better comprehension.
Apart from that the paragraph describing importance of species delimitation and use of barcodes from lines 60-81 is very generalized and basic. It can be improved by avoiding basic details and specifically justifying the importance of species and barcodes.
The in text references used in introduction are quite old it would be better to use updated references. Moreover, the sequence of references throughout the manuscript is not uniform e.g. line 35 should be (Liu et al., 2009; Wang et al., 2009; Wang et al. 2006). Similarly line 59, should be (2004; 2003; 2001). Lines 65, 71, 74,94-95, 101 …and so on.
In the last paragraph of introduction, lines 103-112, it is not clear whether the DNA barcoding approach failed to discriminate the whole genus or just the complex in question of Myricaria.

The figures are clear and nicely described. Tables are quite detailed and well labelled some of the table labels include name of species of Myricaria without being italicized it is requested to recheck all the tables before finalizing. All the raw data has been supplied.

Experimental design

I appreciate the authors for extensively analyzing the data sets and describing in such detail. Their work is highly commendable. However, there are few concerns that need to be addressed.

Firstly, the total sample size and number of data sets, it is difficult to understand how many taxa were used like when you read from lines 115-122 maybe it can be rewritten in a clearer way to help comprehend all sort of audience.

Secondly all the samples were morphologically identified by a single person. I suggest it would be better to consider more than one source to avoid misidentification.

Many softwares are used in the study and the authors have properly mentioned the version and reference however missed for few e.g. line 142, Organellar Genome Draw, line 148, Bowtie2 2.4.5 and SAMtools 1.10, line 149, Geneious 8.1.4, line 156, IQ-TREE, line 309, mVISTA software, line 329, REPuter.

Validity of the findings

The findings are valid and mostly support the previous studies however suggesting to recognize M. bracteata and M. paniculata as two varieties of M. squamosa based on their results is a novel finding.

I suggest to consider rewriting discussion subheading “phylogenetic analysis & species boundaries within M. squamosa complex” specifically line 405, “it was highly supported the incorporation”, line 417, should be species instead of specie. Also from lines 429-438 and even beyond because it is quite difficult to understand what the authors are referring. Like in line 431 “but in conflicted with the former morphological criteria”. Line 437 “two ESUs” this is a new abbreviation not expanded before.

---

## Round 0.2 · Minor Revisions

Please take into account every suggestion by Reviewer 2, that made a thorough review. In addition, in my opinion, the distribution map does not permit the reader to determine the region of distribution, a larger and more detailed map should be included, perhaps a larger map indicating the region and then the detail of distribution. Also in some of the legends, the Maximum Likelihood is not written correctly. The names of the terminals of the cladogram are not readable.

Reviewer 2 has suggested that you cite specific references. You are welcome to add it/them if you believe they are relevant. However, you are not required to include these citations, and if you do not include them, this will not influence my decision.

Reviewer 1 ·

Basic reporting

The revised version of the article “Insights into the phylogenetic relationships and species boundaries in the Myricaria squamosa complex (Tamaricaceae) based on the complete chloroplast genomes” provides new data about the phylogenetic history of the genus Myricaria. The article has greatly improved in comparison to the previous version. The authors have corrected all the suggested aspects. As a consequence, the article has a clear English language, the objectives are clear, and has a more structured layout and high quality figures. Some minor corrections that I have found:
-Every time that a genus or species is first cited in the text, you should include the name of the species written in full (without abbreviation of the genus) and include the authors. Check this aspect throughout the manuscript, because I have found some mistakes (line 92). Besides, the name of the authors should only be included the first time a species or genus is cited (lines 48 and 67).
-Line 293 Remove “using the mVISTA server (Frazer et al., 2004)”. It is already stated in the methodology section.
-Line 419 Change the abbreviation cp by cloroplast
-Line 489 The plural of taxon is taxa. Change it in the sentence.

Experimental design

The experimental design is correct and appropriate to the proposed objectives. The authors have clarified the requested aspects in the revised manuscript.

Validity of the findings

The results of the article provide interesting new data about the phylogenetic history of the genus Myricaria. The authors have improved the reasoning of the text and have included new additional analyses as requested. As a consequence, the conclusions are supported by the data. I have small concerns about two different sentences:
-Lines 466-469 “While the ITS tree in this study (Figure 7B) showed a parallel relationship between M. elegans and the remaining Myricaria species, without valid PP and BS values, it did not provide sufficient evidence for including M. elegans within Myricaria.” Check the phrasing of this sentence, it is a bit weird. Besides, the tree with the marker ITS is not well resolved, probably because this marker does not have enough variability. I would state this aspect in the text.
-Line 480 “For example, M. laxiflora likely originated from hybridization between M. wardii and the M. squamosa complex.” In this sentence, you state that this process is likely. However, I do not think that the genetic data that you provide support this aspect. Maybe you are considering other data, such as morphological aspects, to support this statement? The reticulate gene flow that you suggest in the next sentence is enough to explain the observed tree. Provide a more deep reasoning about this aspect, or delete this sentence.

Reviewer 2 ·

Basic reporting

The article entitled “insights into the phylogenetic relationships and species boundries in the Myricaria squamosal complex (Tamaricaceae) based on the complete chloroplast genomes” will be a good contribution to the journal but there are slight changes that need to be incorporated before the formal submission.
In the abstract, (l. 25,26) the name of the M. squamosa complex is written in both the lines it is better to use it once and replace the second time usage with either “the said complex” or “these unresolved taxa” to avoid repetition. Moreover, the word “underscores” (l. 41) is not appropriate if the study favors/promotes a need for urgent taxonomic revision.
The introduction is much better than before with clear and improved English. Literature is relevant and referenced properly however it would be better if more recent references can be added. The lines 66-69 are not clear and doesn’t clearly depict what the authors want to address it seems as if the sentence is incomplete. In line 92-95 other species names can also be added along with the given example of Ostryopsis intermedia.
I suggest the new sentence in line 131 should be used as new paragraph as it is different from the previous paragraphs. The first objective line 146 “structural features” will be more appropriate. Moreover, I suggest comparative analysis with nrITS could also be added in the second objective as it is used in a heading in methods (l. 196).
The figures are nice and clear but I suggest that there should be a proper figure legend specifically for the figures S1 and S2 both of which share the same heading “Phylogenetic species based on plastome phylogenetic analyses” that is quite generalized and misleading. Similarly figure S3 has no proper legends. These are the main figures of the study giving insights and comparative analysis among different barcodes for species delimitation I suggest they may be included in main text and not as supplementary data.
In figure 7 A/B the legend is quite descriptive and detailed, one thing to notice is that the authors have utilized 39 Myricaria plastome accesstions with 4 used as outgroups as mentioned in results section (l. 350) while 41 samples (l. 375) for nrITS whereas 40 are mentioned in the figure. Moreover, the text size can be increased in the tree of figure 7 figure S3 and colors used specifically for M. elegans and M. laxiflora in figure 7 A/B are not much different I suggest to use a brighter color for proper differentiation.

Experimental design

The experimental design is appropriate and align well with the proposed objectives however I have few suggestions.

First of all, if it’s possible to mention the years in which the sampling was carried out and for how long or repeated over the same seasons this will help validate the points proposed later in discussion section of this study. Moreover, in Sampling it is mentioned that 10 individual samples were taken including M. prostrata (l. 153) however this sample cannot be seen in any of the figures?

Secondly I suggest the plastome feature analysis should be given a single heading and all its supplementary analysis be given as subheadings to the main because it corresponds to the first single objective. For instance, the first two headings of the materials & methods section are fine while the third & fourth heading (l.189 & l. 196) can be placed in end before the last heading (l. 235) while fifth & sixth heading (l. 205 & l. 221) be used as subheadings hence roughly aligning according to the objectives. Similar pattern is observed in results section.

Please expand the of software abbreviations BWA (l.175) & PGA (l. 177). Please add “was used as a reference” (l. 178). Reference of Geneious software is used twice in the text (l. 177 & l. 187) I think if the version is same it can be introduced once.

Regarding the species-specific barcode development in this study which is carried out by using hyper-variable regions that tend to have high species resolution. However, a barcode should have moderate variability to be phylogenetically informative, highly variable and highly conserved both cannot give desired results. Moreover, for intra specificity barcoding gaps can give more insight for such relationships.

Apart from that, it was observed instead of proper references of the software their online links are provided as seen in lines (201) for IRscope, line 209 for EMBOSS, and CodonW. Similarly, line 222 for MISA and line 230 for REPuter. While some software lacked any reference like in line 214 Perl script & line 216 KaKs Calculator 2

It is mentioned under the phylogenetic profiles and species boundary test heading (l. 236-237) that only cases where more than one individual was sampled per species were used it would be better to mention those samples and the ones excluded if any for better comprehension.

In the Bayesian inference analysis as described the MC3 algorithm a variant of MCMC is used but it is not mentioned in the text (l. 252).

Validity of the findings

The findings of the study seem valid and interesting as it includes comparison between whole chloroplast and ITS sequences and has shown different effectiveness of barcodes.

In results section again the main heading for Plastome characteristics can be followed by subheadings including the descriptive analysis of the plastome.

Lines 277-279 can be included in the discussion section instead as a supporting point for discrepancy.

Line 292 figure/table number can be given instead of writing (see plastome phylogenetic results)

Line 296 can also be included in discussion.

Please cite jModeltest and MAFT (l. 350, 351, 378)

The most interesting findings are given in the last two headings of the results.
It’s worth mentioning that according to the figure 7A/B Clade B (cluster v) consists of M. elegans while Clade A (cluster i-iv) grouped remaining Myricaria species not the other way round for both plastome & ITS sequences as mentioned in results (l. 354-355).
It would be better to mention the names of barcodes used combination matrix in lines 397 and 404.
The figures mentioned (l.423) can be shifted in the beginning rather than just before the in text reference.
Expand IGSs and IGRs (l. 425, l. 437) not mentioned earlier.
Can you please provide a reference to support the hybridization between M. wardii and M. squamosa complex (l. 481).
It would be better to further elaborate the point given in lines 536-538. Correct spelling of organismal (l. 551).
Another reference (Channa et al., 2018) can be added with similar argument for M. elegans being a hybrid/intermediate species (l. 463)
Channa, F. N., Shinwari, Z. K. and Ali, S. I. (2018) Phylogeny of Tamaricaceae using psbA-trnH nucleotide sequences. Pak. J. Bot., 50(3): 983-987.

Overall some observations can be drawn from the study:
1. The study provides detailed insight into the features of Myricaria whole chloroplasts but not much difference or uniqueness was observed apart from ycf genes being absent in 3 individuals.
2. Apart from the nrITS, results obtained from the whole plastome, standard cpDNA barcodes and Myricaria-specific barcodes almost aligned in a similar way and none of the interspecies relationships inferred from whole plastome were supported by the ITS sequences. Although no matter how short the standard barcode datasets maybe the effectiveness and the species relationships obtained from them cannot be overlooked and conclusion cannot be made on the whole chloroplast genome alone.
3. M. bracteata and M. paniculata can be considered as varieties of M. squamosa.

Additional comments

The article provides interesting new data regarding Myricaria species and obtained results have been addressed well in the discussion section. I commend the authors for their extensive research. It would be a good contribution to the journal and I think after some minor changes it should be considered for publication.

---

## Round 0.3 · accepted · Accept

Thank you for considering all issues raised in the second round of review. The ms seems now very well organized. My only suggestion, that I made previously is to change to color the local map, but it is O.K.